# Evaluating the Effects of Digital Elevation Models in Landslide Susceptibility Mapping in Rangamati District, Bangladesh

**Yasin Wahid Rabby** [1,*] , **Asif Ishtiaque** [2] **and Md. Shahinoor Rahman** [3]

1   Department of Geography, University of Tennessee, Knoxville, TN 37996, USA
2   School for Environment and Sustainability, University of Michigan, Ann Arbor, MI 48109, USA; asis@umich.edu
3   Department of Earth and Environmental Sciences, New Jersey City University, Jersey City, NJ 07305, USA; mrahman1@njcu.edu
*   Correspondence: yrabby@vols.utk.edu; Tel.: +1-865-4550269

**Abstract:** Digital elevation models (DEMs) are the most obvious data sources in landslide susceptibility assessment. Many landslide casual factors are often generated from DEMs. Most studies on landslide susceptibility assessments rely on freely available DEMs. However, very little is known about the performance of different DEMs with varying spatial resolutions on the accurate assessment of landslide susceptibility. This study compared the performance of four different DEMs including 30 m Advanced Spaceborne Thermal Emission and Reflection Radiometer (ASTER) Global Digital Elevation Model (GDEM), 30–90 m Shuttle Radar Topographic Mission (SRTM), 12.5 m Advanced Land Observation Satellite (ALOS) Phased Array Type L band Synthetic Aperture Radar (PALSAR), and 25 m Survey of Bangladesh (SOB) DEM in landslide susceptibility assessment in the Rangamati district in Bangladesh. This study used three different landslide susceptibility assessment techniques: modified frequency ratio (bivariate model), logistic regression (multivariate model), and random forest (machine-learning model). This study explored two scenarios of landslide susceptibility assessment: using only DEM-derived causal factors and using both DEM-derived factors as well as other common factors. The success and prediction rate curves indicate that the SRTM DEM provides the highest accuracies for the bivariate model in both scenarios. Results also reveal that the ALOS PALSAR DEM shows the best performance in landslide susceptibility mapping using the logistics regression and the random forest models. A relatively finer resolution DEM, the SOB DEM, shows the lowest accuracies compared to other DEMs for all models and scenarios. It can also be noted that the performance of all DEMs except the SOB DEM is close (72%–84%) considering the success and prediction accuracies. Therefore, anyone of the three global DEMs: ASTER, SRTM, and ALOS PALSAR can be used for landslide susceptibility mapping in the study area.

**Keywords:** landslide susceptibility; Bangladesh; digital elevation model; random forest; modified frequency ratio; logistic regression

## 1. Introduction

Local terrain conditions, including terrain relief, hydrology, geology, and land use, are crucial for the assessment of landslide susceptibility [1]. These local features are often termed as "causal factors" [2]. Identifying these causal factors is considered as the steppingstone of landslide susceptibility assessment. Landslide susceptibility represents the likelihood of landslide occurrence in an area. It assumes that future landslides may occur in previous landslide locations where the causal factors already created conducive environments for triggering landslides [2–4]. Several natural and anthropogenic

factors can trigger landslides, thus called "triggering factors," such as volcanic activity, groundwater excavation, prolonged rainfall, rapid snow melting, hill cutting, deforestation, land-use change, etc. [5–8]. Moreover, landslide susceptibility assessment relies on the characteristics of landslide inventory—a detailed register of distribution and characteristics of past landslides [9,10]. Therefore, the success of landslide susceptibility assessment largely depends on the selection of causal factors and the quality of landslide inventories.

Landslide susceptibility can be assessed qualitatively or quantitatively [11]. Qualitative approaches of landslide susceptibility assessment are based on experts' judgments on causal factors. However, the mathematical relationships between landslide locations and casual factors are utilized in quantitative approaches [6]. Frequently, mixed methods and semi-quantitative methods are adopted to process experts' opinions in qualitative assessments. The widely-used methods in this domain are the analytical hierarchy process (AHP) [12], fuzzy logic [13], and GIS-based AHP [14]. In bivariate techniques, each causal factor is divided into a set of classes, and landslide locations are compared with each class. Thus, the bivariate relationship is established between landslide occurrence and one factor-class at a time [4]. The commonly used bivariate methods are frequency ratio, the weight of evidence, fuzzy logic, evidential belief function, and statistical index [3,15]. In contrast, the multivariate statistical methods determine the relationship between landslide occurrence and multiple causal factors. Examples of multivariate methods are logistic regression, adaptive regression spline, general additive models, and simple decision trees [16].

One of the limitations of bivariate and multivariate models is that they are constrained by normality and collinearity assumptions. Compared to these models, machine learning-based models are relatively less limited by these assumptions [17] and, therefore, can consider the nonlinear nature of landslides [18]. Some argue that machine learning-based models such as the random forest, gradient boosting, and support vector machines often outperform both bivariate and multivariate statistical models [19,20]. While the selection of methods is essential, landslide susceptibility assessment also depends on the types and quality of causal factors, mapping unit, and the scale of investigation [21]. Many causal factors are often derived from analyzing satellite imageries and topographic models, including land cover, elevation, slope, aspect, and hill cut [7]. Thus, these derived causal factors are often impacted by the spatial resolution of sources, geometric error, and instrument or sensor type. In landslide susceptibility mapping, digital elevation models (DEMs) often replace topographic maps to derive the most important causal factors (e.g., slope, topography, aspect). Moreover, many developing countries may not have topographic maps. Thus, landslide studies from these countries usually rely on the free of charge DEMs derived from remote sensors.

DEM is the digital representation of the earth's surface. It is widely used in various research areas in which topography plays an important role, such as hydrological modeling, geomorphological analysis, and feature extraction, landslide susceptibility and hazard assessment, erosion susceptibility, and glacier monitoring [22]. DEMs are often generated using data obtained from different remote sensors, including optical imaging sensors, light detection, and ranging (LiDAR), and synthetic aperture radar (SAR) [23]. The qualities of DEM-derived factors often depend on the spatial resolution of DEMs. Therefore, the choice of DEM is important for the assessment of landslide susceptibility [1]. To this day, a few attempts have been made to compare the performance of DEMs with different spatial resolutions in landslide susceptibility assessment. For instance, Dietrich et al. [24] compared different DEMs and found a similarity in performance irrespective of spatial resolution in identifying moderate landslide class (see also [25]). They argued that the resolution of DEM might not be very important to represent the slope failures. Similarly, Tian et al. [26] contended that finer resolution does not essentially lead to higher accuracy in landslide susceptibility assessment (see also [1,27]).

These past studies further indicated that the performance of DEMs is context-dependent meaning that the performance of a DEM in a region may not be assumed to be similar in another region [28–30]. They also argued that DEMs with fine spatial resolution may not necessarily have better performance over coarse resolution DEMs. Therefore, it is an utmost need to have comparative assessments of DEMs

in various contexts. We found that even though some parts of Bangladesh are vulnerable to landslides, no study investigated the relative performance of different DEMs in landslide susceptibility assessments. Against this backdrop, this study contextualized landslide susceptibility in Bangladesh and compared the performance of different DEMs and modeling techniques. The study area is selected from Bangladesh because the hilly southeastern parts of the country encounter landslides almost every year that often claim tens of lives [31–34]. Because of unavailability of LiDAR, the majority of the landslide susceptibility-related studies in Bangladesh used 30m Advanced Spaceborne Thermal Emission and Reflection Radiometer (ASTER) Global Digital Elevation Model (GDEM), 30–90 m Shuttle Radar Topographic Mission (SRTM), and 12.5 m Advanced Land Observation Satellite (ALOS) Phased Array type L-band Synthetic Aperture Radar (PALSAR) to obtain various topographic factors [7,31,32,35]. Survey of Bangladesh (SOB) has developed a DEM (25 m) for the whole of Bangladesh and no study has used this DEM in landslide susceptibility assessment. Therefore, the landslide susceptibility maps in Bangladesh are influenced by the usage of different DEMs and which DEM provides the most accurate susceptibility maps are largely unexplored [36]. As many causal factors are derived from DEM datasets, the selection of appropriate DEM is crucial for landslide susceptibility assessment. This study aims to compare the relative performance of four DEMs: ASTER GDEM (30 m), ALOS PALSAR (12.5 m), SRTM (30 m), and Survey of Bangladesh (SOB) DEM (25 m). This study further aims to compare three quantitative landslide susceptibility assessment techniques: modified frequency ratio (bivariate method), logistic regression (multivariate method), and random forest (machine learning method) [4,8,29,37].

## 2. Study Area

Rangamati, a hilly southeastern district of Bangladesh is selected as the study area because of the regular landslide occurrences in this area. More than 100 people died, and 12,000 families suffered losses due to landslides in this district [38]. Most landslides in Rangamati take place in three upazilas (sub-districts): Rangamati Sadar, Kaptai, and Kawkhali [35]. As such, the study area is narrowed down to these three Upazilas (Figure 1). The combined geographical area of these three sub-districts is 1145 km$^2$ and more than 40% of it is forested [39]. The geology of this area comprises Dhihing, Dupi tila, Girujan clay, Bhuban, Bokabil, and Tipam sandstone (Figure A3f of Appendix C)). The bedrock and soil structure of the areas are not stable, which makes the hills highly prone to landslides [40]. Climatologically, this area falls under a tropical monsoon climate, and the annual average temperature varies from a maximum of 36.5 degrees to a minimum of 12.5 degrees Celsius, and annual rainfall is 2673 mm with mean humidity level 71.6% [39].

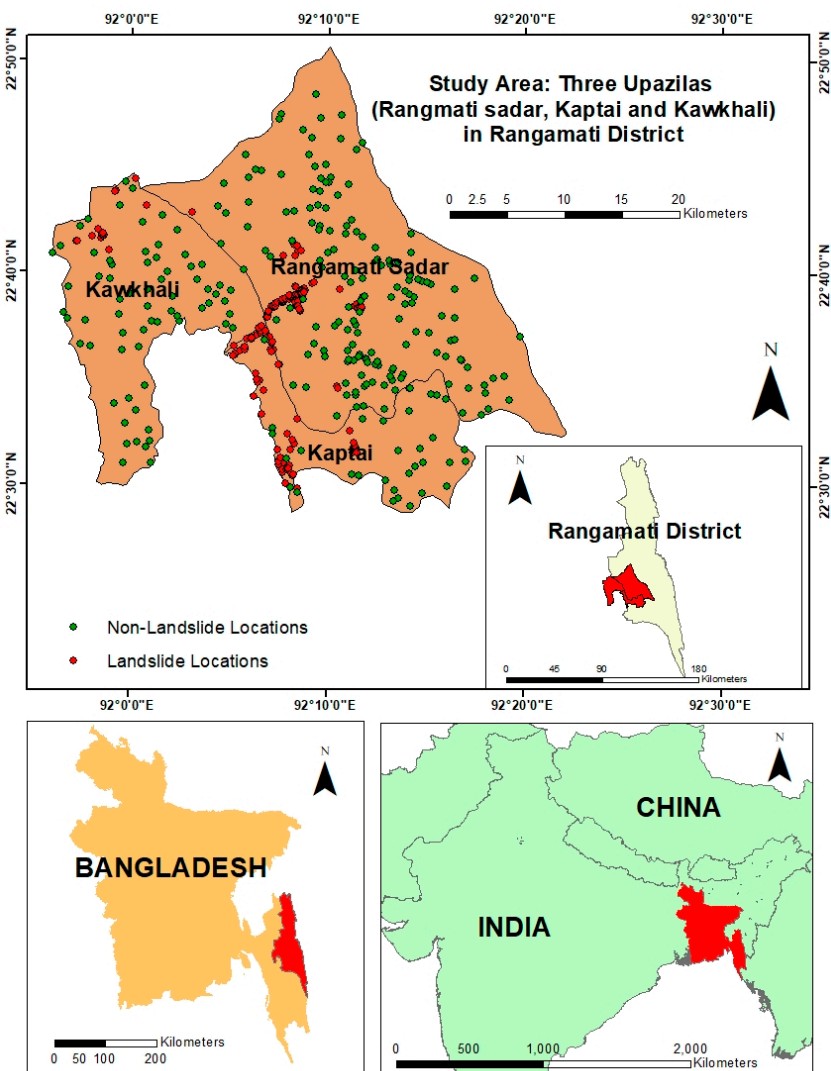

**Figure 1.** Study area and landslide and non-landslide (pseudo-absence points) locations used in modeling.

## 3. Synopsis on Data Utilization

### 3.1. Landslide Inventory

Landslide inventory records different information, including the exact location, size, type, time of occurrence, causalities, trigger, and causes [2]. Rabby and Li [33] prepared and published [34] landslide inventory of the Chittagong hilly area, Bangladesh. They used participatory field mapping to prepare this inventory [41]. In our study, we selected 168 landslides from their inventory as it has been advised to use more than one method in landslide inventory preparation [42], we analyzed available Google Earth images on the Google Earth platform to map more landslides in the study area. We used the method proposed by Rabby and Li [33] for Google Earth mapping and mapped 93 landslides that occurred from January 2001 to January 2019. Therefore, in our study, we used a total of 261 (168 + 93) landslide locations. The mean size of the landslide was 274.2 m$^2$. The smallest and the largest dimensions of the landslides were about 14.6 m$^2$ and 3422.4 m$^2$, respectively.

### 3.2. Landslide Causal Factors

In our study, we used a total of 15 causal factors (Table 1) to produce landslide susceptibility maps (see Appendices A–C). Of these factors, seven factors were derived from DEM: elevation,

slope, plan curvature, profile curvature, topographic wetness index (TWI), stream power index (SPI), and aspect. As we are comparing the performance of different DEMs, we derived each of these factors from four different DEMs: ASTER, SRTM, ALOS PALSAR, and SOB. The rest of the eight factors namely land use/land cover, land use/land cover change, geology, distance from the road networks, distance from the fault lines, distance from the drainage networks, rainfall, and normalized difference vegetation index (NDVI) were collected from different datasets. Maps of different causal factors had different resolutions, but for the convenience of comparison, we kept the 30 m resolution as the standard for landslide susceptibility maps. In the following sub-section, we provide a brief overview of the causal factors that we used in this study. We classify these factors into several classes primarily using Jenks Natural Break method in ArcGIS 10.7, unless otherwise mentioned.

**Table 1.** Data source of the causal factors and resolution.

| Causal Factor | Type | Data Source and Resolution |
|---|---|---|
| Elevation | DEM based | ASTER (30 m), SRTM (30 m), ALOS PALSAR (12.5 m) and SOB (25 m) DEMs |
| Slope | DEM based | ASTER (30 m), SRTM (30 m), ALOS PALSAR (12.5 m) and SOB (25 m) DEMs |
| Aspect | DEM based | ASTER (30 m), SRTM (30 m), ALOS PALSAR (12.5 m) and SOB (25 m) DEMs |
| Plan curvature | DEM based | ASTER (30 m), SRTM (30 m), ALOS PALSAR (12.5 m) and SOB (25 m) DEMs |
| Profile curvature | DEM based | ASTER (30 m), SRTM (30 m), ALOS PALSAR (12.5 m) and SOB (25 m) DEMs |
| Topographic wetness index (TWI) | DEM based | ASTER (30 m), SRTM (30 m), ALOS PALSAR (12.5 m) and SOB (25m) DEMs |
| Stream power index (SPI) | DEM based | ASTER (30 m), SRTM (30 m), ALOS PALSAR (12.5 m) and SOB (25 m) DEMs |
| Rainfall | Other Factors | Bangladesh Meteorological Department (BMD) (1000 m) |
| Distance from the road networks | Other Factors | http://data.gov.bd/dataset/geodash (1000 m) |
| Distance from the drainage networks | Other Factors | http://data.gov.bd/dataset/geodash (1000 m) |
| Distance from the fault lines | Other Factors | Geological Survey of Bangladesh (GSB) (1000 m) |
| Normalized difference vegetation index (NDVI) | Other Factors | Landsat 8 level 2 imagery (30 m) |
| Geology | Other Factors | Geological Survey of Bangladesh (GSB) (1000 m) |
| Land use/land cover | Other Factors | Landsat 8 level 2 imagery (30 m) |
| Land use/land cover change | Other Factors | Landsat 8 level 2 imagery, Landsat 5 imagery (30 m) |

### 3.2.1. Elevation

A change in elevation can bring changes in geomorphology, vegetation, and rate of erosion in an area and thus alters the landslide susceptibility [8]. We derived elevation from ASTER, SRTM, ALOS PALSAR, and SOB DEMs (Figure A1a–d of Appendix A) and divided them into five classes (see Table A1 of Appendix D).

### 3.2.2. Slope

The slope is one of the most critical factors of landslides. Generally, with the increase of slope, shear stress increases, and therefore landslide susceptibility increases [43,44]. Like elevation, we derived slopes from four different DEMs using the slope tool in ArcGIS 10.7 (Figure A1e–h of Appendix A) and divided them into five classes using the Jenks natural break method (see Table A1 of Appendix D). We found different maximum slope values for different DEMs-ASTER (51.89°), SRTM (61.24°), ALOS PALSAR (65.36°), and SOB (46.4°). As these values were different for the same study area, the five classes of slopes (see Table A1 of Appendix D) were different.

### 3.2.3. Aspect

The direction of precipitation, sunlight, and wind depend on aspect, and therefore it has effects on the growth of vegetation, rate of erosion, and thickness of soil [45]. From the DEMs, four aspect maps (Figure A1i–l of Appendix A) were prepared and were divided into ten classes (see Table A1 of Appendix D).

### 3.2.4. Plan Curvature and Profile Curvature

Curvature is the rate of change of slope over time in an area. We used the four different DEMs to produce four plan and profile curvature maps (Plan: Figure A2a–d; Profile: A2e–h of Appendix B). Profile and plan curvatures were divided into three classes: concave, convex, and flat. Among these classes, concave slopes are more prone to landslides because water cannot disperse equally on these slopes [5].

### 3.2.5. Topographic Wetness Index and Stream Power Index

Topographic wetness index (TWI) increases with the decrease of the slope; therefore, it is inversely related to landslide susceptibility [4,43]. Stream power index (SPI) represents the erosion power of streams. SPI is directly related to slopes; in a steeper slope, SPI will be higher, representing more erosion power while in a flat alluvial plain, SPI is low [45]. TWI and SPI maps were derived from four DEMs using Equation (1) and Equation (2) (TWI: Figure A2i–l of Appendix B; SPI: Figure A2m–p of Appendix B)

$$TWI = Ln\left(\frac{A}{tan\alpha}\right) \tag{1}$$

$$SPI = A \times tan\alpha \tag{2}$$

where, A = Area of a specific catchment and $\alpha$ = Slope gradient of the specific area. We divided TWI and SPI into five classes (Table A1 of Appendix D)

### 3.3. Rainfall

The intensity and duration of rainfall controls the initiation of landslides [44]. We used the mean annual rainfall of five weather stations of Bangladesh Meteorological Department (BMD) to prepare the rainfall map using the Kriging interpolation method (Figure A3a of Appendix C). We later divided it into five classes (Table A1 of Appendix D).

### 3.4. Distance-Based Causal Factors

Distance from the road networks, drainage networks, and fault lines were the three distance-based causal factors in this study. We used the Euclidean distance tool in ArcGIS 10.7 to derive the distance of landslides from the targeted features: road, drainage, and fault lines (Figure A3b–d of Appendix C) and divided the distances into five classes (Table A1 of Appendix D). Distance from the road networks is one of the most critical factors. The undercutting of slopes during road construction and the vibration created by vehicles damage the slope stability [43]. Drainage network indicates the zone of erosion in an area, and erosion is indirectly linked with the landslide susceptibility [4]. Fault lines indicate the geomorphological discontinuity in an area. Near the fault lines, the shear strength of rock is minimum. Therefore, areas near to the fault lines are prone to landslides [44].

### 3.5. Normalized Difference Vegetation Index (NDVI)

NDVI indicates the growth of vegetation and biomass of an area [46]. Generally, the probability of the occurrence of the landslide on the naturally vegetated surface is lower than the bare lands [8]. We used Landsat 8 level 2 imagery of 11/10/2017 to prepare the NDVI map (Figure A3e) and divide it into five classes (Table A1 of Appendix D).

*3.6. Geology*

The strength of rock and soil permeability depends on the geology of an area. Therefore, geology has an impact on landslide susceptibility [43]. In this study, we used the geological map (Figure A3f) (1:100,000) of the Bangladesh Geological Survey (BGS). There are eight types of geologic formation found here: Dihing and Dupi tila formation; Bokabil formation; Bhuban formation; Tipam Sandstone; Valley alluvium and colluvium; Dihing formation, Girujan clay, and waterbodies.

*3.7. Land Use land Cover*

Land use/land cover and land use/land cover change are the two most crucial landslide causal factors in the study area [35,47]. Ahmed [31] and Rabby and Li [33] found that the rate of change of land use/land cover in our study area is high compared to other adjacent areas. In this study, we used two Landsat imageries to analyze the land use/land cover change (Landsat 5: date of Acquisition: 24/12/1998; Landsat 8: Acquisition Date: 29/11/2018). We used supervised maximum likelihood classifier to classify the 1998 and 2018 images into four land use classes: bare land, vegetation, built-up, and water bodies (Figure A3g of Appendix C). We later employed post-classification change detection techniques to analyze the land use/land cover changes between 1998 and 2018 (Figure A3h of Appendix C).

## 4. Methodology

To compare the effects of four different DEMs: ASTER, SRTM, ALOS PALSAR, and SOB on landslide susceptibility maps, we used a bivariate method: modified frequency ratio (MFR), a multivariate method: logistic regression (LR) and a machine learning method: random forest (RF). We assessed two scenarios: (a) considering only DEM-based seven causal factors in the models, and (b) considering all 15 causal factors, including the DEM-based factors, in the models. As we used three methods: MFR, LR, and RF on four different DEMs under two scenarios, therefore, the outcome would be twenty-four landslide susceptibility maps. For legibility, we used different acronyms in later sections (see Table 2).

**Table 2.** Acronyms used for different models.

| Model | Factors Considered | DEM | Acronym Used |
|---|---|---|---|
| **Modified frequency ratio** | DEM-based 7 factors | ASTER | MFR_ASTER_DEM |
| | | SRTM | MFR_SRTM_DEM |
| | | ALOS PALSAR | MFR_ALOS_DEM |
| | | SOB | MFR_SOB_DEM |
| | All 15 factors | ASTER | MFR_ASTER |
| | | SRTM | MFR_SRTM |
| | | ALOS PALSAR | MFR_ALOS |
| | | SOB | MFR_SOB |
| **Logistic regression** | DEM-based 7 factors | ASTER | LR_ASTER_DEM |
| | | SRTM | LR_SRTM_DEM |
| | | ALOS PALSAR | LR_ALOS_DEM |
| | | SOB | LR_SOB_DEM |
| | All 15 factors | ASTER | LR_ASTER |
| | | SRTM | LR_SRTM |
| | | ALOS PALSAR | LR_ALOS |
| | | SOB | LR_SOB |
| **Random forest** | DEM-based 7 factors | ASTER | RF_ASTER_DEM |
| | | SRTM | RF_SRTM_DEM |
| | | ALOS PALSAR | RF_ALOS_DEM |
| | | SOB | RF_SOB_DEM |
| | All 15 factors | ASTER | RF_ASTER |
| | | SRTM | RF_SRTM |
| | | ALOS PALSAR | RF_ALOS |
| | | SOB | RF_SOB |

### 4.1. Training and Validation Dataset

We divided the 261 landslide locations randomly into training (75%) and validation (25%) datasets. Bivariate MFR is a one-class classification method where non-landslide locations or absence of landslides are not required [48]. On the other hand, for multivariate LR and machine learning-based RF, the selection of non-landslide locations (pseudo-absence points) is essential [4]. Any place that does not have landslide can be considered as non-landslide. We randomly selected 261 non-landslide locations (Figure 1) (pseudo absence points) from the study area [49]. We split these non-landslide locations into training (196) and validation (65) data sets. In total, we had 392 (196: landslide locations; 196: non-landslide locations) data points for training and 130 (65: landslide locations; 65: non-landslide locations) data points for testing the LR and RF models.

### 4.2. Modified Frequency Ratio (MFR)

MFR is an improved version of the widely used frequency ratio (FR) method [35,50]. Lee and Talib [51] proposed FR, which assesses the spatial relationship between the landslide locations and the landslide causal factors [52]. In the FR method, each of the causal factors must be divided into subclasses or categories; for example, in this study, we split slope into five categories using the Jenks natural break method (Table A1 of Appendix D). We calculated the FR values using Equation (3), and later these FR values of each of the subclasses of causal factors were used in the MFR model.

$$FR_{ij} = \frac{N_{ij}/N}{M_{ij}/M} \tag{3}$$

where $FR_{ij}$ = Frequency Ratio of jth Subclass of Factor i

$N_{ij}$ = Total area of the landslide pixels within the jth subclass of factor i

$N$ = Total area of landslide pixels in the study area

$M_{ij}$ = Total area of the pixels in the jth subclass of factor i

$M$ = Total area of the study area

FR > 1 means association of landslides with that subclass. In other words, there is a probability of occurrences of landslides in that subclass. FR < 1 means no association [53].

For calculating the MFR, then we normalized the FRs using Equation (4).

$$Rf_{ij} = \frac{FR_{ij}}{\sum FR_i} \tag{4}$$

where $Rf_{ij}$ = Relative frequency of jth subclass of factor i

$FR_{ij}$ = Frequency ratio of the jth subclass of factor i

$\sum FR_i$ = Sum of the frequency ratios of factor i

Later, we calculated the prediction rate (PR) using Equation (5). In the FR model, the overall contribution of causal factors to the occurrence of landslides is not measured. Only the subclass wise contribution is measured [6,50]. In MFR, we can measure the overall contribution because PR indicates overall association of a causal factor. The lowest value of PR is 1 and the higher the PR value the stronger is the association of causal factor with the landslides [6].

$$PR_i = \frac{(MaxRf_i - MinRf_i)}{(MaxRf_i - MinRf_i)\min} \tag{5}$$

where $PR_i$ = Prediction rate of factor i

$MaxRf_i$ = Maximum relative frequency of factor i

$MinRf_i$ = Minimum relative frequency of factor i

$(MaxRf_i - MinRf_i)_{\min}$ = Lowest difference between maximum and minimum relative frequency of all the factors

To calculate the landslide susceptibility ondex (LSI) and to produce the landslide susceptibility maps Equation (6) was used

$$LSI = \sum_{i=1}^{n} Rf_{ij} \times PR_i \qquad (6)$$

where LSI = Landslide susceptibility index

$Rf_{ij}$ = Relative frequency of jth subclass of factor i

$PR_i$ = Prediction rate of factor i

For landslide susceptibility mapping, we used the reclassify tool in ArcGIS 10.7 to reclassify the categories of causal factors according to the Rf values. Later, we multiplied each of the reclassified raster layers with the prediction rates and summed up to produce the final landslide susceptibility maps.

### 4.3. Logistic Regression (LR)

Logistic regression (LR) is one of the most widely-used multivariate statistical methods in landslide susceptibility mapping [4,6,54–56]. An LR model predicts the presence of landslides using the binary landslide data (presence and absence of landslides or landslides and non-landslides) and their relationship with the landslide causal factors [56,57]. Here, landslide and non-landslide locations are dependent variables, and causal factors are independent variables. These independent variables can be numerical or categorical [4]. Equation (7) is used in LR model

$$\text{Logit } (Y) = ß1X1 + ß2X2 + ß3X3 + \dots\dots\dots\dots\dots\dots\dots\dots\dots\dots\dots + ßiXi + e \qquad (7)$$

where Y = The presence of landslides

$X_i$ = ith Causal factor

$ß_i$ = Regression coefficient of the ith causal factor

e = Error

We used Equation (8) to determine the probability.

$$P = \frac{expY}{1 + expY} \qquad (8)$$

We used the R software environment for the forward stepwise LR method. We multiplied the raster layers of the statistically significant causal factors with the coefficients and summed up using Equation (7) in the R software environment. Finally, we used Equation (8) to produce the landslide susceptibility maps.

### 4.4. Random Forest Classification (RF)

The random forest method was developed by Breiman [58] and is an ensemble learning method [19]. Lately, the use of the RF method for landslide susceptibility mapping has increased due to its high performance in predicting landslide locations [37,59].

This method uses bootstrapping techniques to generate a bunch of classification trees based on subsets of observations [27]. There is high variance among the individual trees, and therefore classification based on a single tree is unstable and prone to overfitting [37]. Random forest is improved over commonly used tree-based methods, such as a decision tree or bagged tree because it decorrelates the trees. RF uses ensembles of trees and lets each tree define the class membership, and finally, the respective class is assigned based on the highest votes [27,37]. Since the bootstrapping method is used, a set of data is not used in the model training stage and this set of data is known as out-of-bag (OOB) [27]. These OOB data are used to calculate the mean decrease of accuracy and Gini coefficient [37]. The accuracy and Gini coefficient are used in variable selection and ranking [19]. They also provide the statistical weights or variable importance of each of the predictors used in the model [27].

There are several advantages of using RF methods, such as rescaling and transformation of data are not essential; missing data and outliers can be ignored [60]. Moreover, it can deal with both

numerical and categorical data, and the use of a dummy variable is not required [37]. In this study, we developed the Random Forest model in the R software environment using the "randomForest" package [61].

### 4.5. Multicollinearity Diagnostics

In the LR model, multicollinearity can bring inaccuracies in variance and unsuitability in estimates. On the other hand, in the RF model, it can affect the variable importance [62]. Therefore, multicollinearity diagnostics: variance inflation factor (VIF) and tolerance were used before using the causal factors in LR and RF models. Since VIF values were <10 and tolerance were <0.3, causal factors were independent and were used in these two models [63]

### 4.6. Model Validation and Comparison

Success and prediction rate curves were used to test the performance of the susceptibility models. The training data set was used for the success rate curve, and the validation data set was used for the prediction rate curve [6]. "The area under the curve" (AUC) of success rate indicates how well the model fits the training data while the AUC of prediction rate suggests how well the model will predict future landslides [15,64]. AUC value ranges from 0–1 or 0%–100% and it can be grouped into the following categories: 0.50–0.60 (fail); 0.60–0.70 (poor); 0.70–0.80 (fair); 0.80–0.90 (good), and 0.90–1.00 (excellent) [53]. We also used two non-parametric tests: Friedman and Wilcoxon Signed Rank test to assess whether there are any significant differences in performances between the susceptibility [65–68]. Friedman's test is used for multiple comparisons. This test determines whether there is any significant difference in performance in multiple models [29], while the Wilcoxon Signed Rank test is used for pairwise comparison of susceptibility models and, therefore, can indicate which models are significantly different [68].

However, statistical performance assessments such as success and prediction rate curves cannot show the level of agreement among the models. Therefore, we used convergent validation through the coverage based cross-comparison [69,70]. The MFR gives landslide susceptibility index while LR and RF provide the probability of landslides. We reclassified them into five susceptibility zones: very low, low moderate, high, and very high using the Jenks natural break method. Later, we used the raster calculator in the ArcGIS platform to subtract the reclassified model from one another. The outcome can be any integer, but only "zero" will indicate the areas which were classified into the same susceptibility zones by two compared models. We calculated the percentage of area under this "zero" class and this percentage indicates the spatial convergence or agreement between two compared models [69].

## 5. Results

### 5.1. Susceptibility Assessment Using Modified Frequency Ratio (MFR)

We found that the prediction rates (PRs) of the slope, aspect, and elevation derived from three DEMs (ASTER, SRTM, and ALOS PALSAR) are similar. The PRs for these three factors are around 2.25, 1.0, and 3.0, respectively (see Table A1 of Appendix D). Compared to these three DEMs, the SOB DEM showed different PRs. Our analysis revealed that for slope, aspect, and elevation, the SOB DEM-based PRs are 1.79, 2.37, and 2.41, respectively. We further found that with the increase of slope, the probability of landslide increases. The relatively safer zones are in areas below 8° slope where frequency ratio (FR) < 1. For ASTER, SRTM, and ALOS PALSAR DEMs, we found that the slope class 14–23° had the highest probability of landslides. However, for SOB DEM, the highest probability of landslide is in the 8–14° slope class. In the case of TWI, we found that ALOS PALSAR DEM has the highest PR (4.30) followed by ASTER DEM (PR = 3.19) and SOB DEM (PR = 1.61). For all the four DEMs, the probability of landslides was higher in areas where TWI is less than 6. The SPIs derived from four DEMs have lower PRs compared to other causal factors and the class-wise weight (FR values)

showed the same sort of pattern. We further found that for plan curvature, ALOS PALSAR has the highest PR (4.30), while for profile curvature, SRTM has the highest PR (4.68).

In general, we found no specific pattern of FR and PR values for these seven factors. We observed that causal factors derived from either ALOS PALSAR or SRTM have higher PR values, and SOB has the lowest PR among the four DEMs. The causal factors derived from different DEMs did not have a significant impact on the FR and PR values of MFR. This finding is similar to the findings of Chang et al. [29]. Most of the topographic factors are the first derivative of the DEMs other than the TWI and SPI. Thus, for TWI, DEMs have more substantial impacts than other factors. It is because there can be a small difference among the DEMs, and when the second derivative is used, these become pronounced [29].

The class-wise FR values of the eight factors that are not derived from the DEMs are generally similar; however, the PR values are different (see Table A1 of Appendix D). The causal factors derived from the DEMs played a crucial role in determining the PR values for these eight factors. For ASTER (0.141), SRTM (0.134), and ALOS PALSAR (0.139), the lowest difference was between the maximum and minimum relative frequency of aspect. While for SOB (0.160), it was SPI. Since these values are slightly different, it affected the PR values.

Landslide Susceptibility Maps (MFR)

We produced Landslide Susceptibility Indices (LSIs) of the four MFR models using Equation (6) for DEM-based causal factors. The LSI of MFR_ASTER_DEM ranged from 994.6 to 8388.1. The LSIs for MFR_SRTM_DEM LSIs for MFR_SRTM_DEM; MFR_ALOS_DEM and MFR_SOB_DEM ranged from 591.4 to 9458.8 and 527.8 to 10056.5 and 638.8 to 6130.0, respectively. LSI does not have a unit. It is the product of relative frequency and prediction rate Equation (6), and both of these do not have units. The greater the LSI, the greater is the landslide susceptibility, and the smaller the LSI value, the lower is the susceptibility [35,44]. The ranges of LSIs indicate that MFR_SRTM_DEM and MFR_ALOS_DEM models had a comparatively broader range than the rest of the two models. This happened because of the variable FR and PR values. For ASTER DEM, the highest PR value was for plan curvature, while for other DEMs it was for profile curvature (see Table A1 of Appendix D). For all DEM-based factors, SOB had the lowest PRs among the four models and therefore it affected the LSIs. Later, we used the same Equation (6) to produce four MFR models based on 15 causal factors. The LSI of MFR_ASTER DEM ranged from 1613.00 to 20370.10. The LSIs for MFR_SRTM, MFR_ALOS, and SOB DEMs ranged from 1314.40 to 22300.34 and 1234.95 to 22180.24 and 1995.7 to 17316.9, respectively. The LSIs of MFR_SRTM and MFR_ALOS DEMs have a comparatively broader range than the rest of the two models. The highest PR value for ASTER was 6.25 and for SOB, it was 5.64 (Table A1 of Appendix D) for distance to the road network, while for SRTM and ALOS PALSAR, it was 6.61 and 6.38, respectively. For other causal factors (Table A1 of Appendix D), SOB had the lowest PRs among the four, and therefore LSI was the lowest. As mentioned earlier, the FR values varied for seven topographic factors derived from four different DEMs, and these factors had impacts on the PR of eight common factors. Ultimately these variations defined the LSI of the susceptibility maps.

Since different models had different LSIs, Rescale by Function tool in ArcGIS was used to normalize the LSIs into a 0.0–1.0 scale. Later, we used the Jenks natural break method to classify the normalized LSIs into five susceptibility zones: very low, low, moderate, high, and very high. Generally, the spatial appearances of the landslide susceptibility maps have similarities with the map of causal factors that have a higher contribution to landslides. In this study, this contribution is shown by PR and FR values (see Appendices A–C). We found that the spatial appearance of seven causal factors derived from SOB was different from the spatial appearance of seven causal factors derived from ASTER, SRTM, and ALOS PALSAR. ASTER, SRTM and ALOS PALSAR based susceptibility maps (Figure 2a–c) show a comparatively lesser percentage of area as very low or low susceptibility zones than the SOB-based landslide susceptibility map (Figure 2d). However, the SRTM based map shows comparatively more areas as high and very high susceptibility zones that the other maps. We found when all factors were

considered, areas near to the road network are highly susceptible to landslides (Figure 3a–d), because the PR values of the distance from the road networks were the highest.

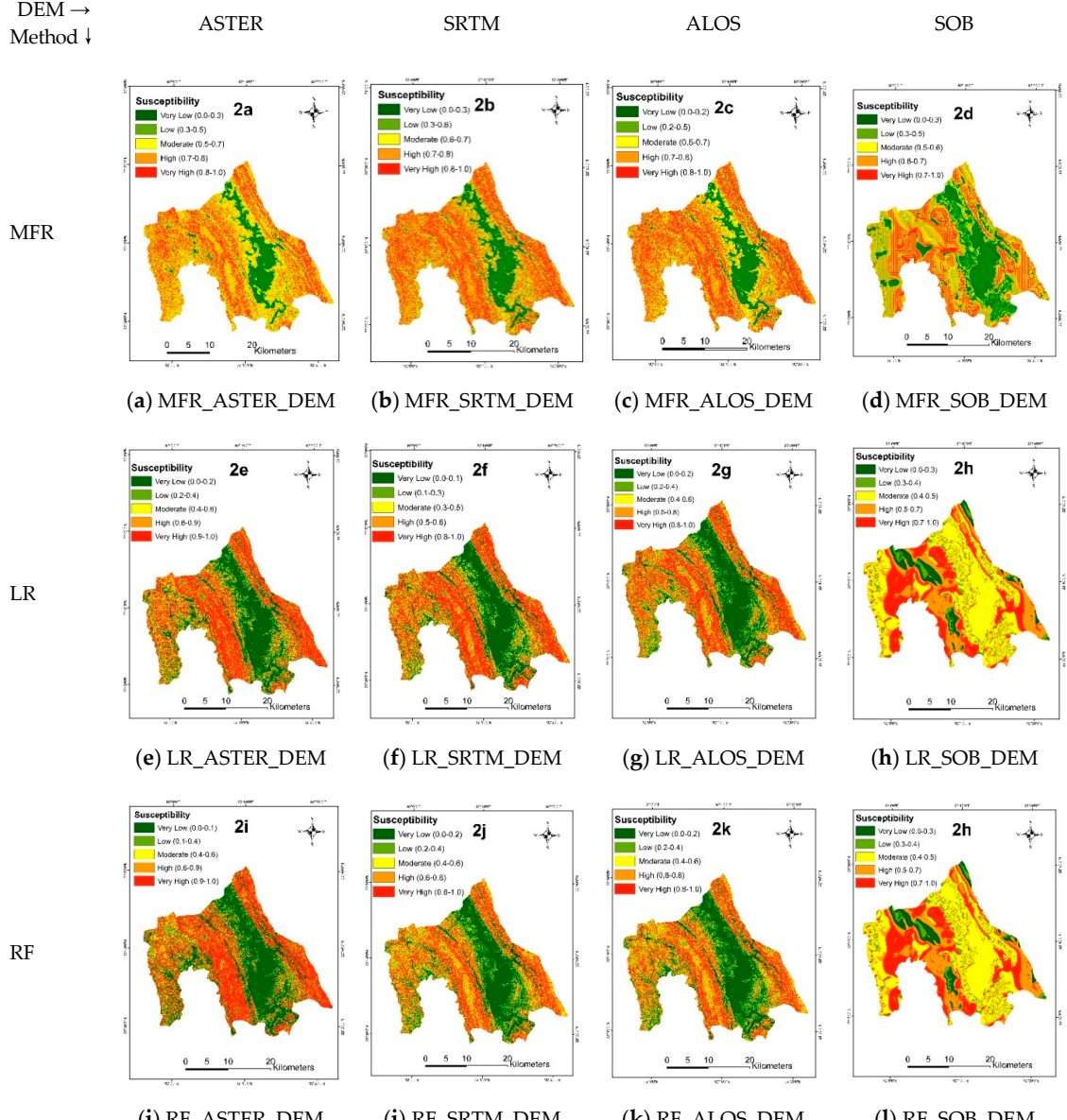

**Figure 2.** Landslide susceptibility maps produced using Modified Frequency Ratio (MFR), Logistic Regression (LR) and Random Forest (RF) models (seven DEM-based factors).

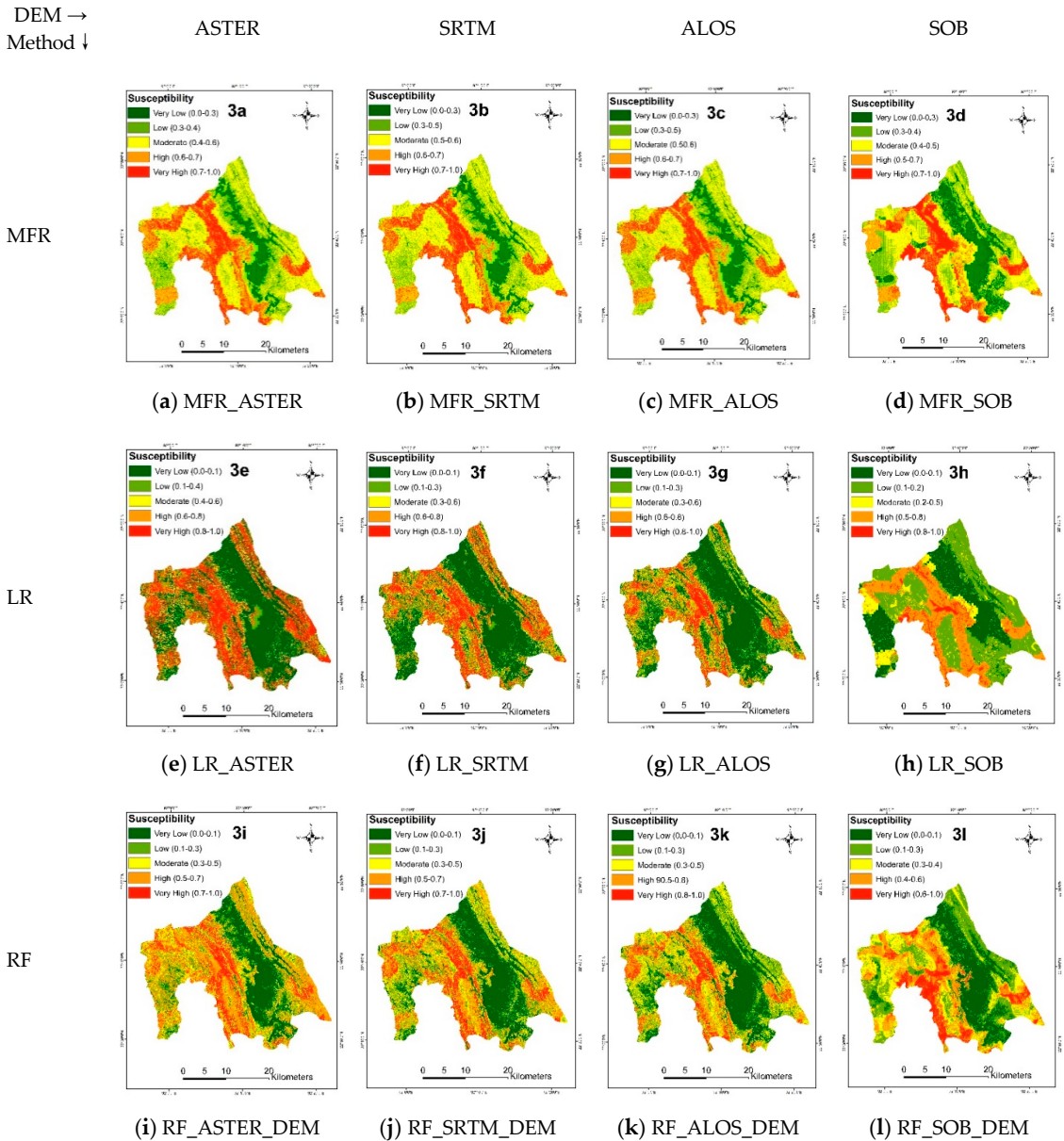

**Figure 3.** Landslide susceptibility maps produced using MFR, LR, and RF models (all 15 factors).

*5.2. Susceptibility Assessment using Logistic Regression (LR)*

For the DEM-based factors, the LR model detected two to five statistically significant factors (see Table A2 of Appendix E). Elevation and slope were the two common statistically significant causal factors for ASTER, SRTM, and ALOS PALSAR based models. Since the ALOS PALSAR based model, the highest number of causal factors was chosen, DEM had the highest impact on the landslide susceptibility map. Odds ratio (Table A2 of Appendix E) shows that slope was the most important factor of landslides for ASTER and SRTM based models, while aspect came out as the most crucial factor for ALOS PALSAR and SOB based models.

When all (15) causal factors were used, we found a total of four to eight statistically significant causal factors (see Table A2 of Appendix E). Slope, elevation, SPI, and aspect were the significant DEM-based causal factors for LR_SRTM and LR_ALOS based models. For LR_SOB, aspect was the statistically significant DEM-based causal factor. When 15 causal factors were used, the model assessed the interaction of DEM-based factors with the common eight factors. Therefore, when only the DEM-based causal factors were used, some factors came out statistically significant (e.g., SPI for

LR_ASTER_DEM). When all the factors were used in the LR_ASTER based model, SPI came out insignificant. For LR_SRTM and LR_ALOS, three DEM-based causal factors were detected as statistically significant; therefore, for these two landslide susceptibility maps, DEM would have more impact than the LR_ASTER and LR_SOB susceptibility maps.

Landslide Susceptibility Maps (LR)

We used the Jenks natural break method to classify the probability of landslides into five zones: very low, low, moderate, high, and very high. The spatial appearances of the LR_SOB_DEM model (Figure 2h) have a different appearance than the other three maps. LR_SRTM_DEM (Figure 2f) and LR_ALOS_DEM (Figure 2g) have an almost identical spatial appearance. While in LR_ASTER_DEM models, the slope had a higher coefficient (ß = 0.26) (Table A2 of Appendix E) than the SRTM (ß = 0.14) and ALOS PALSAR ((ß = 0.05). Therefore, in the LR_ASTER_DEM map, the areas with steeper slopes, mainly in the mid-north and mid-south of the study area, were classified as high to very highly susceptible. In LR_SRTM_DEM and LR_ALOS_DEM maps, these areas were classified either as moderate or high susceptibility zones. In LR_SOB_DEM, only elevation and aspect were two significant factors. Therefore, the susceptibility map took the shape of the map of these two factors (see Appendix A).

When all (15) causal factors were used, the spatial appearance of LR_SOB (Figure 3h) was different from the other three maps (Figure 3e–g). LR_SOB map was influenced by the distance from the fault lines, distance from the road networks, and land use/land cover. Although aspect was a significant factor, the coefficient value of aspect was similar to other significant factors, and distance from the fault lines (ß = 1.07) (Table A2 of Appendix E) had a higher coefficient value than aspect. Therefore, most of the study area was classified as low or very low susceptibility zones. In the LR_ASTER model (Figure 3e), the slope had the highest coefficient (ß = 0.31) (Table A2 of Appendix E), and therefore, areas with steeper slopes were classified as high or very high susceptibility zones. But in LR_SRTM and LR_ALOS models, the slope had comparatively lower coefficient values than ASTER. As a result, some areas were classified as moderate susceptibility zones in these two maps (Figure 3f–g). In LR_SRTM and LR_ALOS susceptibility maps, common factors such as distance from the road networks and fault lines, land use/land cover, and land use/land cover change did not have higher coefficient values than the DEM-based causal factors. That is why, unlike LR_SOB, the spatial appearance of the susceptibility maps did not follow the appearance of the maps of common factors.

*5.3. Susceptibility Assessment using Random Forest (RF)*

RF_ASTER_DEM, RF_SRTM_DEM, and RF_ALOS_DEM models detected (Figure 4a) slope and RF_SOB_DEM model identified the aspect as the most critical factor. When all 15 factors were used in the models, RF_ASTER and RF_SRTM (Figure 4b) detected slope and the rest of the two models detected distance from the road network as the most important causal factor. For RF_ASTER, RF_SRTM, and RF_ALOS, DEM-based factors such as elevation, TWI, and aspect had higher importance in the models than the common factors. But in the RF_SOB model DEM-based factors had less importance than the common factors. There is no similarity among the models in detecting the importance of DEM-based causal factors. For example, In RF_SOB, the slope was ranked as one of the least important factors, but for other models, it was ranked as the most important factor (Figure 4b).

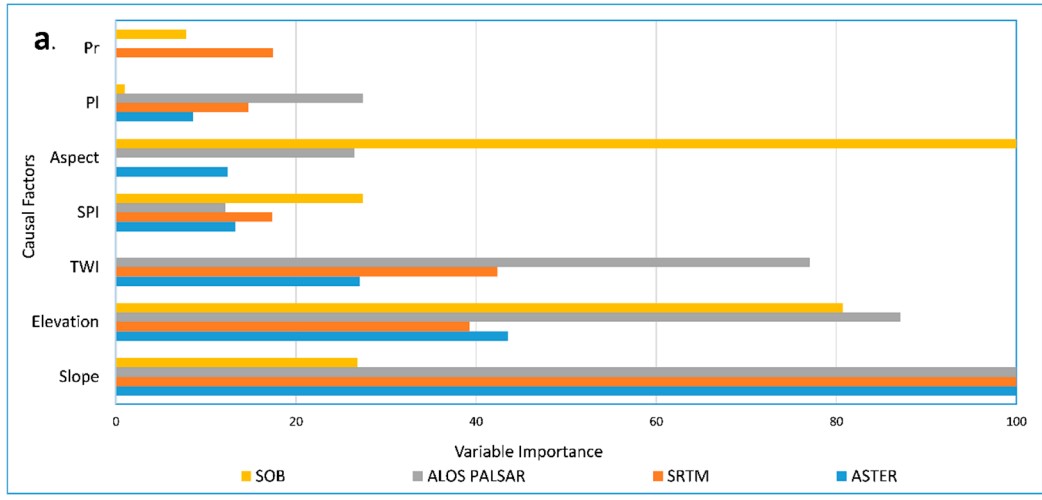

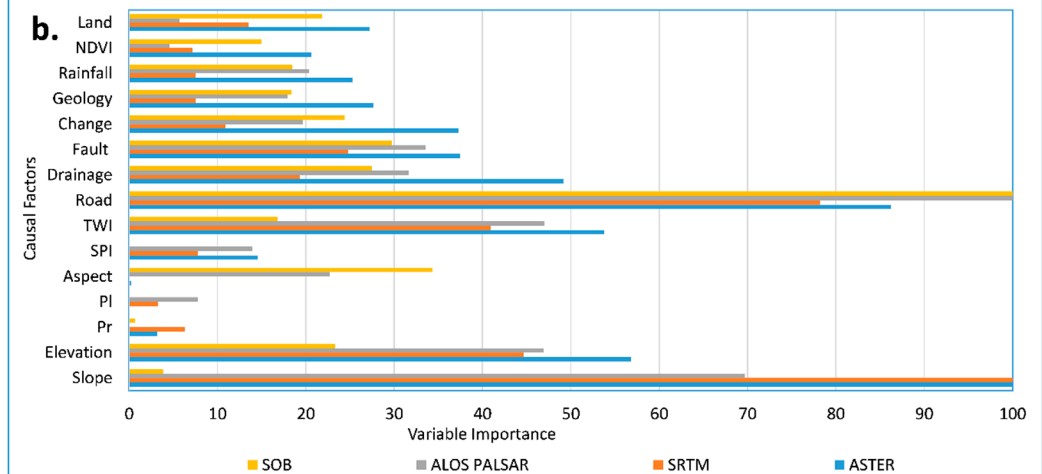

**Figure 4.** Variable importance plots of random forest model: (**a**) Digital Elevation Model (DEM)based causal factor (7) used in the models. (**b**) All (15) factors used in the models. Pl=plan curvature; Pr= profile curvature; LC= land use/land cover; LLC= land use/land cover change; DF= distance from the fault lines; DD= distance from the drainage networks; DR= distance from the road networks.

Landslide Susceptibility Maps (RF)

Like MFR and LR, we used the same method to classify the probability of landslides into five susceptibility zones. The spatial appearance of the RF_SOB_DEM susceptibility map (Figure 2l) was different from the susceptibility maps of the other three models (Figure 2i–k). For, RF_ASTER_DEM (Figure 2i) areas in the mid-north to mid-south were classified as high or very high susceptibility zones. While in RF_SOB_DEM and RF_ALOS_DEM the same areas were classified as moderate susceptibility zones. In RF_ASTER_DEM, slope was the most critical factor. Similarly, in RF_SRTM_DEM and RF_ALOS_DEM slope was the most crucial factor, but in these two models, the contribution of the slope (Figure 4a) to the model is lesser than the RF_ASTER_DEM model. In RF_ASTER_DEM, the difference of variable importance between slope and other factors was comparatively higher than the other models, the effect of slope on the susceptibility map was visible.

RF_ASTER; RF_SRTM, RF_ALOS; and RF_SOB models (Figure 3i–l), spatial appearances were different from each other. Since in the RF_SOB model, distance from the road networks was the most crucial factor, areas near to roads were classified as high or very high susceptibility zones. Distance from the road network was not ranked as the most critical factor in the other three models.

### 5.4. Validation and Comparison of Landslide Susceptibility Maps

#### 5.4.1. Success and Prediction Rate Curves

##### DEM-Based Causal Factors

When only DEM-based seven causal factors were used for MFR models, among all DEMs the MFR_SRTM_DEM model gave the superior performance for both success (AUC = 80.73%) and prediction (AUC = 77.37%) rate curves (Figure 5a,b). The MFR_SOB_DEM model appears to perform the weakest in assessing success and prediction. The AUCs of success rate curves (Figure 5a) showed that MFR_SRTM_DEM falls under the good category while MFR_ASTER_DEM and MFR_ALOS_DEM fall under the fair category. But MFR_SOB_DEM falls under the poor category. AUCs of prediction rate curves (Figure 5b) show that all models other than the MFR_SOB_DEM gave fair performances.

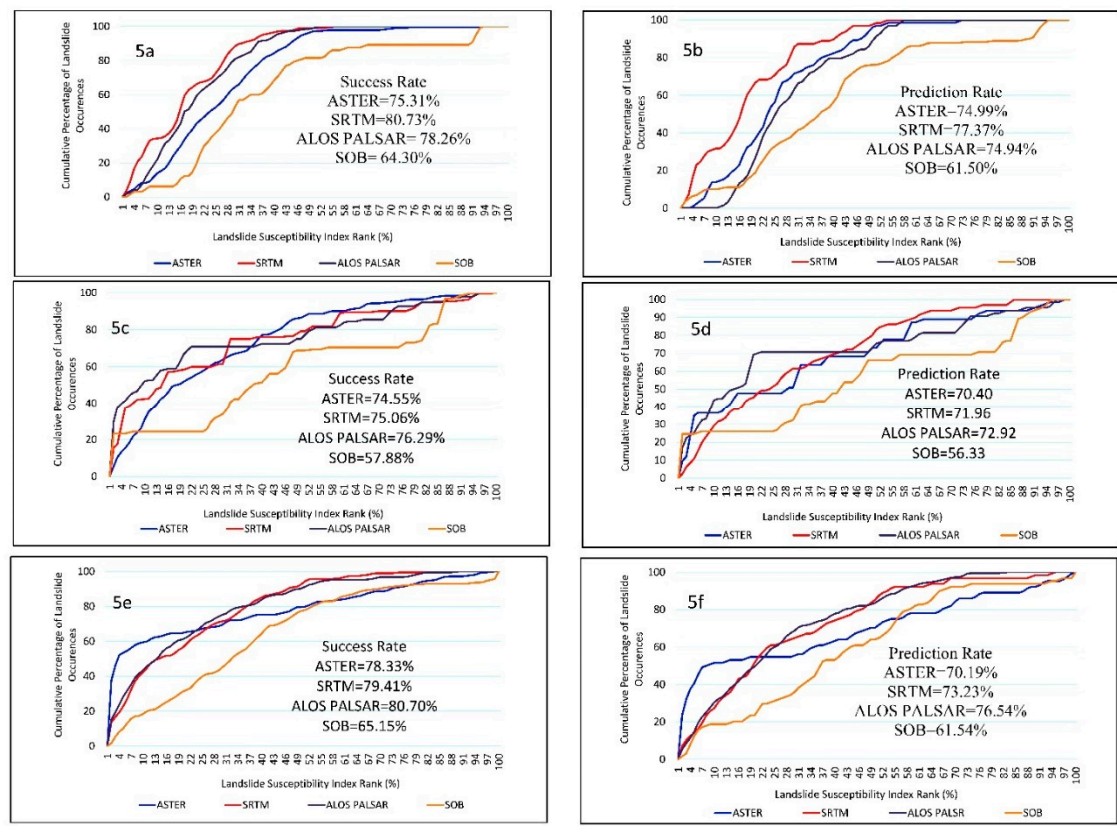

**Figure 5.** Success and prediction rate curves (seven DEM-based factors): (**a**) MFR (success); (**b**) MFR (prediction); (**c**) LR (success); (**d**) LR (prediction); (**e**) RF (success); (**f**) RF (prediction).

For LR, LR_ALOS_DEM outperformed the other three models (Figure 5c,d). LR_SOB_DEM presented the weakest performance among the four models and thus fell under the fail category. The AUCs of success and prediction rates (Figure 5c,d) show that the other three models are under the fair category. For RF models, we got similar results as the LR model. RF_ALOS_DEM outperformed other models, and RF_SOB_DEM was the least accurate model. RF_SRTM_ALOS_DEM and RF_SRTM_DEM gave an almost similar performance.

##### All Causal Factors

We found that when all 15 causal factors are used, different models showed variable performances. For the MFR model, the use of 15 causal factors decreases the predictive performance on an average by 5% of landslide susceptibility maps based on ASTER, SRTM, and ALOS PALSAR (see Figure 6a,b).

However, for the SOB based model, it increased the performance by 3%. It indicates that for bivariate models, DEM-based causal factors can give better prediction performance and use of non-DEM-based factors can reduce the accuracy. Inclusion of more DEM-based causal factors and more landslide locations may increase the accuracy of the models in the study area.

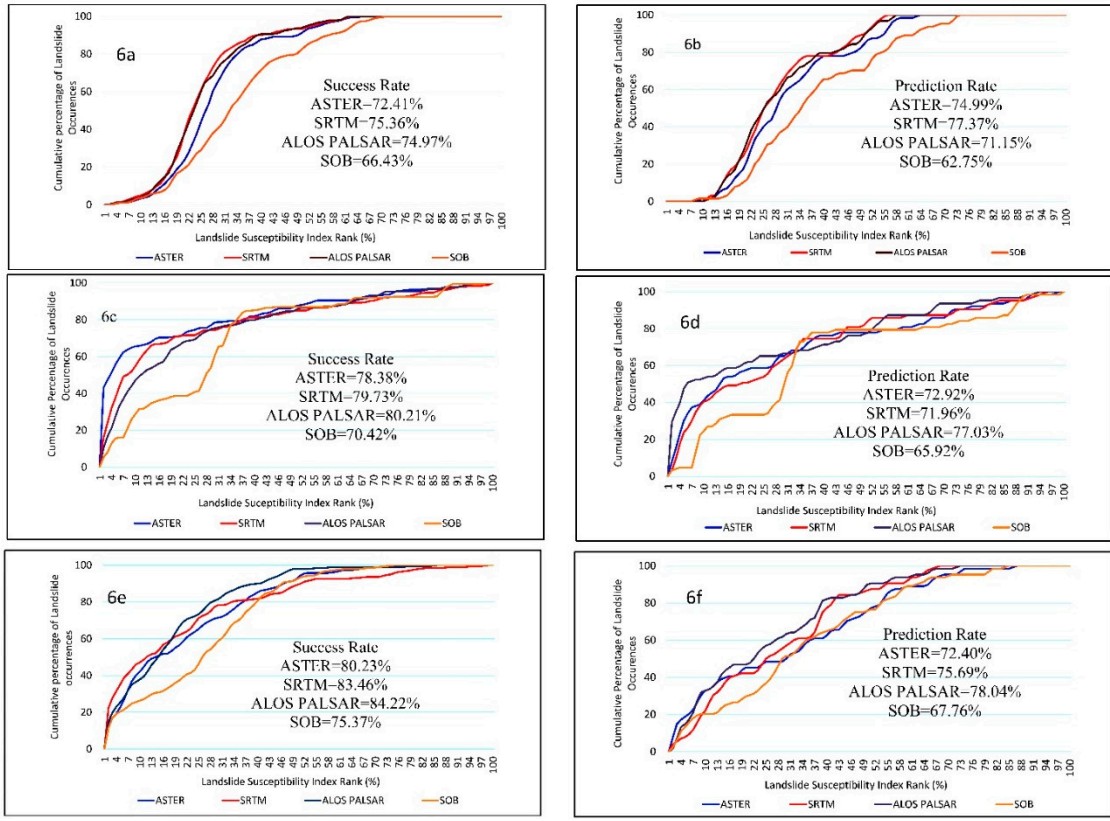

**Figure 6.** Success and prediction rate curves (15 factors): (**a**) MFR (success); (**b**) MFR (prediction); (**c**) LR (success); (**d**) LR (prediction); (**e**) RF (success); (**f**) RF (prediction).

For the LR model, the use of 15 factors increased the accuracy of the model. For three global DEMs success rates increased by around 5.0% but for SOB it increased by 21.7%. On the other hand, prediction rates showed the same trend as for three global DEMs the increase of performance was around 3.5% but for SOB it was 17.0%. It proves that the use of common factors increased the accuracy substantially for SOB DEM.

Like the LR model, for RF models, the use of 15 causal factors increased the accuracy of the model. For ASTER and SRTM the increase of the success rate was around 7%. For ALOS PALSAR the success rate increased by 12.4%. Here again, SOB had the highest increase (22.6%) in success rate. The increase in prediction rate was not as high as the success rate. For three global DEMs prediction rates increased by around 2%–3% and for SOB the prediction rate increased by 10.1%. Machine learning algorithms such as the random forest learn the behavior or the training data. Therefore, the increase of success rate due to inclusion of new variables was high. Since machine learning algorithms learn the behavior of the training data it fails to predict the validation or unknown data [60]. Therefore, in our study, the increase of prediction rate is around 50% lower than the increase of prediction rate for RF models.

5.4.2. Spatial Comparison of Landslide Susceptibility Maps

Spatial convergence indicates how much area is classified into same susceptibility zones. When seven DEM-based factors were used in the MFR model, MFR_SOB_DEM had 30% of spatial convergence while the remaining DEMs had around 40% (Table 3). As we discussed before, the landslide

susceptibility maps of MFR_SOB_DEM has a different spatial appearance (Figure 2d) than the rest of the three susceptibility maps and these results (Table 2) support the previous discussion of our study. For the LR models, Table 3 shows a similar trend. Spatial convergences of the LR_ASTER_DEM, LR_SRTM_DEM, and LR_ALOS_DEM were around 44%, while the LR_SOB_DEM showed approximately 19% of spatial convergence. For the RF models, Table 3 shows the similarities with the findings of MFR and LR.

**Table 3.** Spatial comparison and convergence analysis of landslide susceptibility maps.

| Factors Used | Method Mapping | DEM | ASTER (%) | SRTM (%) | ALOS PALSAR (%) | SOB (%) |
|---|---|---|---|---|---|---|
| **DEM-based factors** | MFR | ASTER (%) | 100.00 | 41.18 | 46.54 | 29.37 |
| | | SRTM (%) | 41.18 | 100.00 | 42.54 | 31.26 |
| | | ALOS PALSAR (%) | 46.53 | 42.54 | 100.00 | 30.83 |
| | | SOB (%) | 29.37 | 31.26 | 30.83 | 100.00 |
| | LR | ASTER (%) | 100.00 | 44.34 | 47.50 | 18.33 |
| | | SRTM (%) | 44.34 | 100.00 | 48.59 | 18.84 |
| | | ALOS PALSAR (%) | 47.50 | 48.59 | 100.00 | 19.06 |
| | | SOB (%) | 18.33 | 18.84 | 19.06 | 100.00 |
| | Random forest | ASTER (%) | 100.00 | 47.73 | 47.07 | 28.24 |
| | | SRTM (%) | 47.73 | 100.00 | 54.46 | 31.43 |
| | | ALOS PALSAR (%) | 47.07 | 54.46 | 100.00 | 31.46 |
| | | SOB (%) | 28.24 | 31.43 | 31.46 | 100.00 |
| **All factors** | MFR | ASTER (%) | 100.00 | 71.44 | 68.89 | 55.71 |
| | | SRTM (%) | 71.44 | 100.00 | 77.41 | 54.07 |
| | | ALOS PALSAR (%) | 68.89 | 77.41 | 100.00 | 51.68 |
| | | SOB (%) | 55.71 | 54.07 | 51.68 | 100.00 |
| | LR | ASTER (%) | 100.00 | 52.01 | 51.11 | 30.89 |
| | | SRTM (%) | 52.01 | 100.00 | 55.56 | 36.46 |
| | | ALOS PALSAR (%) | 51.11 | 55.56 | 100.00 | 36.07 |
| | | SOB (%) | 30.89 | 36.46 | 36.07 | 100.00 |
| | Random forest | ASTER (%) | 100.00 | 51.55 | 49.28 | 42.21 |
| | | SRTM (%) | 51.55 | 100.00 | 62.63 | 44.36 |
| | | ALOS PALSAR (%) | 49.28 | 62.63 | 100.00 | 47.02 |
| | | SOB (%) | 42.21 | 44.36 | 47.02 | 100.00 |

When all factors were considered for modeling, spatial convergence between the DEMs (Table 3) increased around 40% for MFR models. While for the LR and RF models, the spatial convergence was approximately 25% and 12%, respectively. In the MFR model, all causal factors were used, while in the LR model, significant causal factors were used and in the RF model, 2–3 causal factors, for example, profile and plan curvatures had comparatively low or no variable importance in the model.

The results of Friedman tests (Table 4) show that in both the scenarios (a. seven DEM-based causal factors used, and b. 15 causal factors used) for MFR and RF models $P < 0.05$. It means at least one of the landslide susceptibilities models had significantly different performance than the rest of the models. While for LR models, when seven DEM-based causal factors were used, at least one of the models was statistically different in performance than the rest of the models. When all factors were used in LR models, there was no statistically significant difference in performance between the models. Wilcoxon signed-rank test conducted the pairwise comparison. The results (Table 4) show that in scenario two, the performances of landslide susceptibility maps produced using SRTM and ALOS PALSAR did not have a statistically significant difference. Other than that, all the performances of the MFR based landslide susceptibility maps were statistically ($\alpha = 0.008$ after Bonferroni correction) different from each other. When seven causal factors were used for LR models, the performance of SOB based models was significantly different (Table 5) from all other models. But when all factors are used, these differences become insignificant. It indicates that eight common factors overshadow the effect of SOB based causal factors. In the MFR model, it did not happen since it did not consider the interaction of causal factors.

**Table 4.** Result of the Friedman Test for landslide susceptibility maps.

| Factors Used | Methods | DEM | Mean Rank | Chi-Square | *P*-Value |
|---|---|---|---|---|---|
| DEM-based factors | MFR | ASTER | 2.29 | 135.71 | 0.00 * |
| | | SRTM | 2.94 | | |
| | | ALOS PALSAR | 2.68 | | |
| | | SOB | 2.10 | | |
| | LR | ASTER | 2.56 | 45.24 | 0.00 * |
| | | SRTM | 2.55 | | |
| | | ALOS PALSAR | 2.19 | | |
| | | SOB | 2.70 | | |
| | RF | ASTER | 2.61 | 44.24 | 0.00 * |
| | | SRTM | 2.28 | | |
| | | ALOS PALSAR | 2.74 | | |
| | | SOB | 2.37 | | |
| All factors | MFR | ASTER | 2.61 | 208.54 | 0.00 * |
| | | SRTM | 2.28 | | |
| | | ALOS PALSAR | 2.74 | | |
| | | SOB | 2.37 | | |
| | LR | ASTER | 2.37 | 8.63 | 0.05 |
| | | SRTM | 2.90 | | |
| | | ALOS PALSAR | 2.84 | | |
| | | SOB | 1.89 | | |
| | RF | ASTER | 2.76 | 33.79 | 0.00 * |
| | | SRTM | 2.43 | | |
| | | ALOS PALSAR | 2.49 | | |
| | | SOB | 2.32 | | |

In the case of RF models, RF_ALOS_DEM (Table 5) was statistically different from the other models. But when all causal factors were used the difference of performance became insignificant for RF_SRTM model. RF used a more complex algorithm than the LR and MFR models. Therefore, the Wilcoxon Signed-Rank test gave different results for RF models than MFR and LR models.

**Table 5.** Comparison of landslide susceptibility maps based on three different DEMs using Wilcoxon signed-rank test.

| Factors Used | Methods | Pairwise Comparison | Z-Statistics | *P*-Value |
|---|---|---|---|---|
| DEM-based factors | MFR | ASTER-SRTM | −8.82 | 0.00 * |
| | | ASTER-ALOS PALSAR | −4.81 | 0.00 * |
| | | ASTER-SOB | −6.81 | 0.00 * |
| | | SRTM-ALOS PALSAR | −5.26 | 0.00 * |
| | | SRTM-SOB | −10.48 | 0.00 * |
| | | ALOS PALSAR-SOB | −8.60 | 0.00 * |
| | LR | ASTER-SRTM | 1.11 | 0.27 |
| | | ASTER-ALOS PALSAR | −1.49 | 0.14 |
| | | ASTER-SOB | −3.62 | 0.00 * |
| | | SRTM-ALOS PALSAR | −2.55 | 0.01 |
| | | SRTM-SOB | −3.04 | 0.00 * |
| | | ALOS PALSAR-SOB | −3.71 | 0.00 * |
| | RF | ASTER-SRTM | −0.42 | 0.68 |
| | | ASTER-ALOS PALSAR | −3.37 | 0.00 * |
| | | ASTER-SOB | −7.06 | 0.48 |
| | | SRTM-ALOS PALSAR | −7.37 | 0.00 * |
| | | SRTM-SOB | −1.79 | 0.73 |
| | | ALOS PALSAR-SOB | −6.14 | 0.00 * |
| All factors | MFR | ASTER-SRTM | −7.91 | 0.00 * |
| | | ASTER-ALOS PALSAR | −6.11 | 0.00 * |
| | | ASTER-SOB | −10.58 | 0.00 * |
| | | SRTM-ALOS PALSAR | −2.11 | 0.04 |
| | | SRTM-SOB | −12.84 | 0.00 * |
| | | ALOS PALSAR-SOB | −11.56 | 0.00 * |
| | LR | ASTER-SRTM | −2.37 | 0.02 |
| | | ASTER-ALOS PALSAR | −1.59 | 0.11 |
| | | ASTER-SOB | −0.49 | 0.63 |
| | | SRTM-ALOS PALSAR | −0.53 | 0.60 |
| | | SRTM-SOB | −0.24 | 0.81 |
| | | ALOS PALSAR-SOB | −0.08 | 0.94 |
| | RF | ASTER-SRTM | −2.66 | 0.01 |
| | | ASTER-ALOS PALSAR | −0.38 | 0.71 |
| | | ASTER-SOB | −4.64 | 0.00 * |
| | | SRTM-ALOS PALSAR | −1.85 | 0.07 |
| | | SRTM-SOB | −1.96 | 0.05 |
| | | ALOS PALSAR-SOB | −4.18 | 0.00 * |

* = Significant after Bonferroni correction ($P < 0.008$).

## 6. Discussion

This paper evaluates the suitability of three available global DEMs: ASTER, SRTM, and ALOS PALSAR and a local DEM: SOB for landslide susceptibility mapping in Rangamati district, Bangladesh. Causal factors derived from ASTER and ALOS PALSAR DEM have been used in landslide susceptibility mapping in different parts of the Chittagong hilly areas, Bangladesh [7,9,31–35]. Since the study areas of these studies were different, we could not compare them to find out which DEM gives the best accuracy in the prediction of landslide susceptibility [36]. Our study showed that three global DEMs outperformed the local DEM in all three landslide modeling scenarios.

In the first scenario, only DEM-based causal factors were used in modeling and for MFR models, MFR_SRTM_DEM outperformed the other three models. The difference of AUCs of both the success and prediction rate curves between _SRTM_DEM and ALOS_DEM was around 3%, indicating a similarity in predictions. For the processing of ALOS PALSAR DEM, SRTM GL1 data is used for radiometric correction [71]. Therefore, the quality of ALOS PALSAR DEM depends on the quality of SRTM DEM. SOB DEM-based MFR did not show a good performance for the study area and the prediction performance can be improved when a more representative landslide inventory with more landslide locations is used. We found that the ASTER DEM-based MFR model showed a weaker performance than the other two open-source global DEMs. Our result is consistent with other studies that utilized ASTER DEM [29,72,73]. It may happen because ASTER DEM contains many artifacts such as the presence of peaks in the flat terrain, and it ultimately affects the landslide susceptibility map [29]. The poor performance of the SOB DEM can be attributed to the interpolation methods that were used to extrapolate elevations from the available spot heights in the hilly parts of Bangladesh [74]. It affected the accuracy and quality of DEM in the hilly parts of Bangladesh. On the other hand, for LR and RF, ALOS PALSAR based models outperformed the rest of the models. Here, again the difference between LR_ALOS_DEM and LR_SRTM_DEM was low for success and prediction. In all cases, SOB based models gave the worst performance and causal factors derived from SOB DEM cannot explain the landslide susceptibility of the study area. For example: In LR_SOB_DEM, the LR model used two significant causal factors: elevation and aspect (Table A2 of Appendix E). The low coefficient (ß) values of these two factors indicate that these two causal factors cannot adequately explain the landslide susceptibility of the study area.

In the second, scenario, for MFR models, the use of 15 causal factors increased the prediction accuracy for MFR_SOB. But for the other three models, it reduced accuracy. It indicates that causal factors derived from three global DEMs were capable enough to explain the landslide susceptibility of the study area. MFR is a bivariate model, and it does not consider the interaction of the causal factors. Moreover, unlike LR and RF models, it does not require non-landslide (pseudo absence point) in modeling [44]. When all causal factors were used, PRs of some of the common causal factors were comparatively higher than the PRs of the DEM-based causal factors. For example, PRs of distance from the road networks were around 6.00 (Table A1 of Appendix D), while PRs of the DEM-based causal factors ranged from 1.00–4.43. Therefore, the higher PRs of the common causal factors overshadowed the DEM-based factors [6,35,44]. On the other hand, the quality of the SOB DEM was poor and was not capable of explaining the landslide susceptibility of the study area. That is why, in the MFR_SOB model, the prediction accuracy increased, but the increase was very low. For LR and RF models, in second scenarios, the prediction accuracy increased by 2%–3% for global DEM-based models, while for SOB based models, it exceeded 10%. Both LR and RF models consider the interaction of causal factors, and therefore, in these two models, the effects of common factors on prediction performance were revealed better than the MFR models.

The findings of this study have similarities with other research where the suitability of different global and local DEMs was evaluated for landslide susceptibility mapping [26–28]. In most of the studies, ASTER DEM-based landslide susceptibility maps were outperformed by the other global DEM-based landslide susceptibility maps. On the other hand, local DEM-based landslide susceptibility maps have better prediction accuracy than the global DEMs [1,29]. In these studies, local DEMs

were mainly light detection and ranging (LiDAR)-based DEMs, which generally have very high spatial resolution compared to global DEMs [29]. SOB DEM was prepared under the project titles "Improvement of Digital Mapping System of Survey of Bangladesh", where the main aim was to prepare a 1:5000 scale DEM of major cities in Bangladesh. This SOB project prepared DEM for the hilly areas of Bangladesh, including our study area using the local spot heights. Different interpolation methods were used to prepare 25m DEM from these local spot heights [73]. Therefore, in hilly areas of Bangladesh, the quality of SOB DEM is not good enough, and the application of this DEM in geomorphological studies such as landslide susceptibility mapping can give questionable results similar to what we got in our study. Our study also pointed out the importance of the preparation of very high-resolution DEMs using LiDAR for the hilly areas of Bangladesh. It will help in various geomorphological studies, including landslide susceptibility mapping. As we did not find a substantial difference among global DEMs, any global DEM can be utilized for landslide susceptibility mapping in Bangladesh in the absence of very high-resolution DEMs.

For an in-depth study, we utilized non-parametric tests. Chang et al. [30] used non-parametric tests to detect a significant difference in performance in landslide susceptibility maps prepared using different DEM-derived causal factors. Their study did not find any significant difference in performance for two machine learning methods: RF and support vector machines. In our research, we did not see any significant difference in performance for RF and LR based models. But for the MFR model, we found a significant difference in performance. It indicates that the effect of DEM-based causal factors on the performance of bivariate models is more than on the multivariate and machine learning models. Thus, we suggest using multivariate and machine learning methods and any one of the global DEMs for landslide susceptibility mapping in Bangladesh.

## 7. Conclusions

This paper assesses the effects of the DEM-derived causal factors on the landslide susceptibility maps produced using the bivariate (e.g., MFR), multivariate (e.g., LR), and machine learning (e.g., RF) models. In this study, we tested two scenarios: a. susceptibility assessment with only seven DEM-based causal factors; b. inclusion of other 8 causal factors along with DEM-derived factors. The success and prediction rate curves showed SRTM DEM outperformed under an MFR model in both scenarios. Our analysis revealed that ALOS PALSAR DEM provided the best prediction accuracy while using both LR and RF models in landslide susceptibility mapping. For all models and scenarios, the SOB DEM does not perform well compared to other DEMs.

The prediction accuracies of landslide susceptibility mapping using only DEM-derived factors is low compared to the utilization of all casual factors. Although SOB DEM has a poor performance in susceptibility assessment with only a DEM-derived factor, the accuracy is significantly improved when other non-DEM-based factors are added. Therefore, we argue that causal factors derived from the SOB DEM have limited influence in landslide susceptibility assessment for the study area. Besides, for the LR and RF models, the use of all the causal factors increased the prediction performance. Spatial convergence analysis showed that three global DEM-based models have similar accuracies and performed far better than SOB DEM-based models. Therefore, we recommend that SOB DEM should not be used for landslide susceptibility assessment in Bangladesh. Although ASTER DEM-based models showed the weakest performance among three global DEMs, the difference of performance was negligible. Therefore, we recommend any one of these three global DEMs can be utilized for landslide susceptibility assessment in Bangladesh.

Our study also highlights that, for the MFR model, DEM had the highest impact on the accuracy of landslide susceptibility assessment as Wilcoxon rank tests showed that the performance of susceptibility maps was significantly different. For the LR and RF models, the effect of DEM was less significant. We contend that while using the bivariate model, we must be careful about the quality of DEM. Two scenarios in this study helped to understand the impact of DEMs in landslide susceptibility assessment. Moreover, we used multiple metrics to evaluate the accuracies of susceptibility assessment

including AUC, spatial convergence analysis, and non-parametric test. Although the use of one performance measure is a common practice, the utilization of various measures helped this research to understand the impacts of DEM and makes the conclusion robust.

The main limitation of our study is that we cannot ascertain if adding more DEM-derived factors such as terrain roughness could improve the result. The impacts of DEM on landslide susceptibility maps are not universal and may vary from place to place. Therefore, we cannot conclude that a specific DEM can be better than others. Advanced machine learning and deep learning methods can be used to check whether these algorithms can reduce the differences in prediction performance of landslide susceptibility maps prepared using different DEM-derived causal factors.

**Author Contributions:** Conceptualization, Y.W.R; methodology, Y.W.R.; M.S.R.; analysis, Y.W.R.; writing, Y.W.R. A.I.; M.S.R.; review and editing, A.I. and M.S.R. All authors have read and agreed to the published version of the manuscript.

**Funding:** This research received no external funding

**Acknowledgments:** The authors thank Bayes Ahmed (Lecturer, Institute for Risk and Disaster Reduction, University College London) for his assistance in obtaining some datasets.

**Conflicts of Interest:** The authors declare no conflict of interest.

## Appendix A

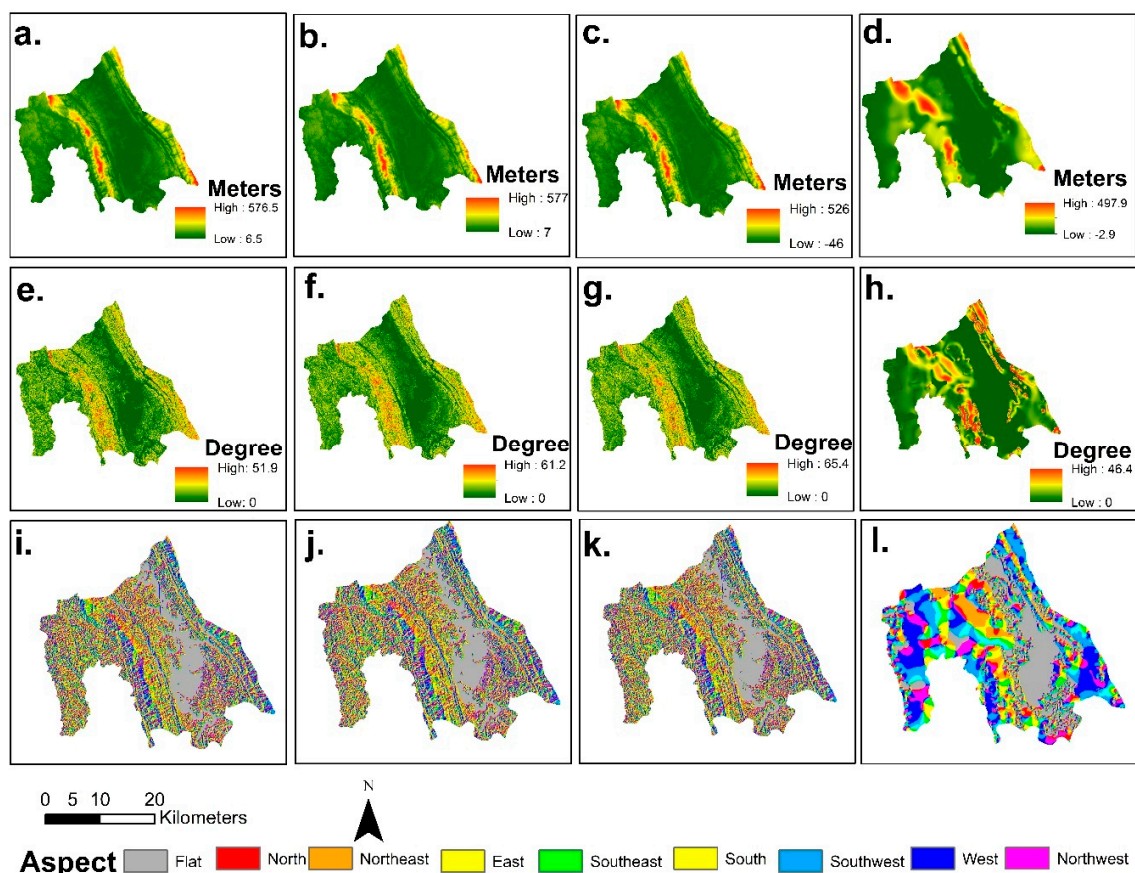

**Figure A1.** Landslide Causal Factors: (**a**) Elevation (ASTER); (**b**) Elevation (SRTM); (**c**) Elevation (ALOS PALSAR); (**d**) Elevation (SOB); (**e**) Slope (ASTER); (**f**) Slope (SRTM); (**g**) Slope (ALOS PALSAR); (**h**) Slope (SOB); (**i**) Aspect (ASTER); (**j**) Aspect (SRTM); (**k**) Aspect (ALOS PALSAR); (**l**) Aspect (SOB).

## Appendix B

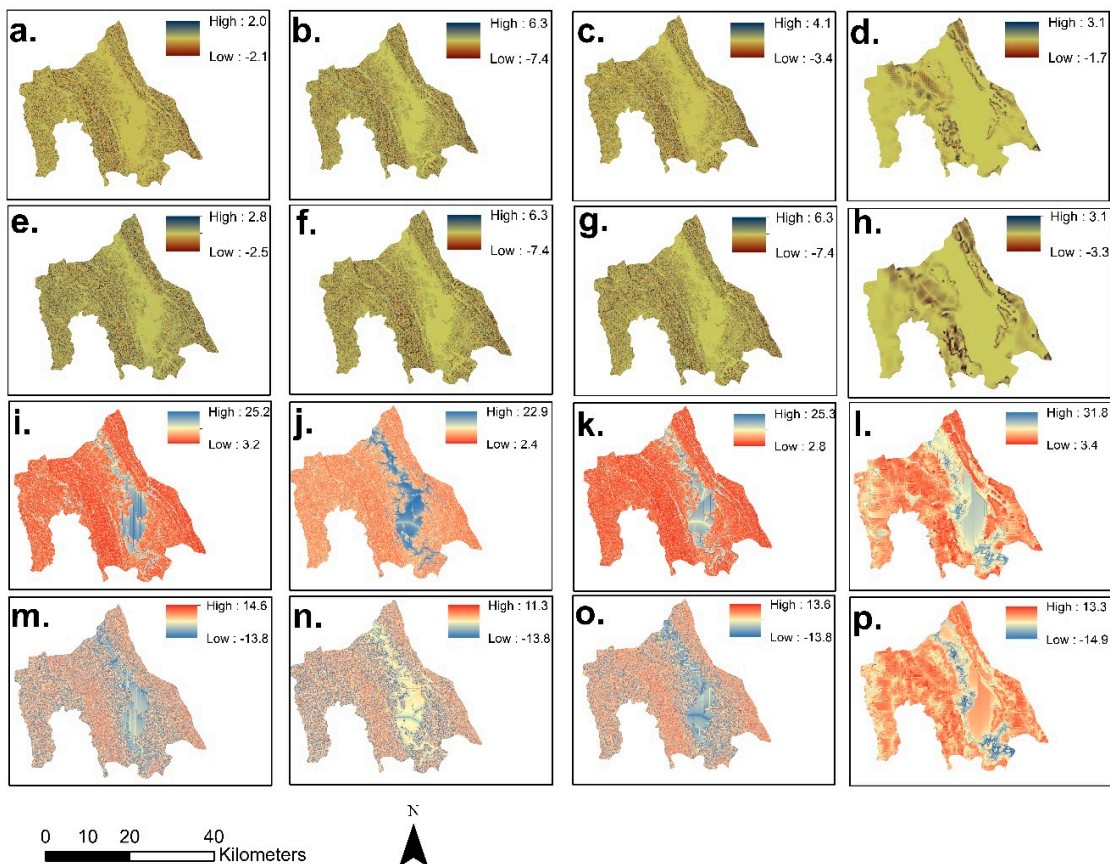

**Figure A2.** Landslide Causal Factors: (**a**) Plan Curvature (ASTER); (**b**) Plan Curvature (SRTM); (**c**) Plan Curvature (ALOS PALSAR); (**d**) Plane Curvature (SOB); (**e**) Profile Curvature (ASTER); (**f**) Profile Curvature (SRTM); (**g**) Profile Curvature (ALOS PALSAR); (**h**) Profile Curvature (SOB); (**i**) TWI (ASTER); (**j**) TWI (SRTM); (**k**) TWI (ALOS PALSAR); (**l**) TWI (SOB); (**m**) SPI (ASTER); (**n**) SPI (SRTM); (**o**) SPI (ALOS PALSAR); (**p**) SPI (SOB).

## Appendix C

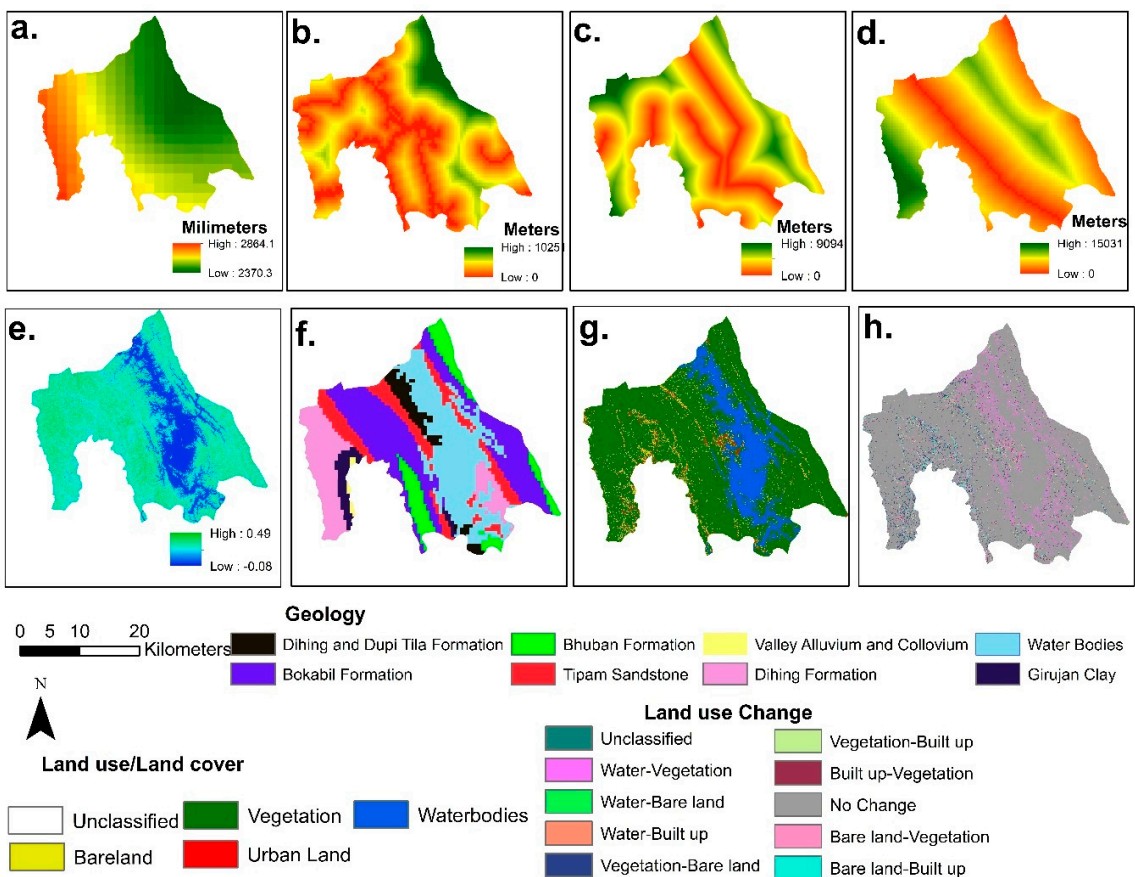

**Figure A3.** Landslide Causal Factors: (**a**) Rainfall; (**b**) Distance from the Road Networks; (**c**) Distance from the Drainage Networks; (**d**) Distance from the Fault Lines; (**e**) NDVI; (**f**) Geology; (**g**) Land Use / Land Cover; (**h**) Land Use/ Land Cover Change.

## Appendix D

**Table A1.** Spatial relationship between Causal Factors and Landslides.ASTER DEM; B= SRTM DEM; C= ALOS PALSAR DEM D= SOB DEM.

| Causal Factors | Classes | Area (%), $\frac{Aij}{Atotal}$ | Landslides (%), $\frac{Nij}{Ntotal}$ | FR, $\frac{Nij/Ntotal}{Aij/Atotal}$ | RF, $\frac{FRij}{\sum FRi}$ | MinRF | MaxRF | (MaxRF - MinRF) | PR |
|---|---|---|---|---|---|---|---|---|---|
| | Flat | 0.00 | 0.00 | 0.00 | 0.00 | | | | |
| | North | 16.7 | 4.17 | 0.25 | 0.03 | | | | |
| | Northeast | 11.5 | 11.46 | 1.00 | 0.11 | | | | |
| | East | 11.3 | 17.71 | 1.57 | 0.17 | | | | |
| **Aspect (ASTER)** | Southeast | 10.2 | 13.54 | 1.33 | 0.14 | 0.03 | 0.17 | 0.14 | 1.00 |
| | South | 10.4 | 13.02 | 1.25 | 0.13 | | | | |
| | Southwest | 13.1 | 18.23 | 1.39 | 0.15 | | | | |
| | West | 12.1 | 7.81 | 0.65 | 0.07 | | | | |
| | Northwest | 10.1 | 9.90 | 0.98 | 0.11 | | | | |
| | North | 4.6 | 4.17 | 0.91 | 0.10 | | | | |

**Table A1.** *Cont.*

| Causal Factors | Classes | Area (%), $\frac{Aij}{Atotal}$ | Landslides (%), $\frac{Nij}{Ntotal}$ | FR, $\frac{Nij/Ntotal}{Aij/Atotal}$ | RF, $\frac{FRij}{\sum FRi}$ | MinRF | MaxRF | (MaxRF - MinRF) | PR |
|---|---|---|---|---|---|---|---|---|---|
| Aspect (SRTM) | Flat | 0.00 | 0.00 | 0.00 | 0.00 | 0.03 | 0.16 | 0.13 | 1.00 |
| | North | 17.40 | 4.69 | 0.27 | 0.03 | | | | |
| | Northeast | 10.99 | 10.42 | 0.95 | 0.10 | | | | |
| | East | 11.19 | 16.67 | 1.49 | 0.16 | | | | |
| | Southeast | 10.31 | 10.94 | 1.06 | 0.11 | | | | |
| | South | 10.45 | 13.02 | 1.25 | 0.13 | | | | |
| | Southwest | 13.45 | 20.31 | 1.51 | 0.16 | | | | |
| | West | 12.31 | 11.98 | 0.97 | 0.10 | | | | |
| | Northwest | 9.65 | 7.81 | 0.81 | 0.09 | | | | |
| | North | 4.26 | 4.17 | 0.98 | 0.11 | | | | |
| Aspect (ALOS PALSAR) | Flat | 0.00 | 0.00 | 0.00 | 0.00 | 0.03 | 0.17 | 0.14 | 1.00 |
| | North | 17.81 | 4.69 | 0.26 | 0.03 | | | | |
| | Northeast | 10.98 | 10.42 | 0.95 | 0.10 | | | | |
| | East | 10.71 | 16.67 | 1.56 | 0.17 | | | | |
| | Southeast | 10.71 | 10.94 | 1.02 | 0.11 | | | | |
| | South | 10.35 | 13.02 | 1.26 | 0.13 | | | | |
| | Southwest | 13.39 | 20.31 | 1.52 | 0.16 | | | | |
| | West | 11.84 | 11.98 | 1.01 | 0.11 | | | | |
| | Northwest | 9.92 | 7.81 | 0.79 | 0.08 | | | | |
| | North | 4.29 | 4.17 | 0.97 | 0.10 | | | | |
| Aspect (SOB) | Flat | 22.99 | 13.76 | 0.60 | 0.05 | 0.03 | 0.40 | 0.37 | 2.37 |
| | North | 8.59 | 3.17 | 0.37 | 0.03 | | | | |
| | Northeast | 8.73 | 5.29 | 0.61 | 0.05 | | | | |
| | East | 7.25 | 3.17 | 0.44 | 0.04 | | | | |
| | Southeast | 6.14 | 29.10 | 4.74 | 0.40 | | | | |
| | South | 7.73 | 6.35 | 0.82 | 0.07 | | | | |
| | Southwest | 12.35 | 11.11 | 0.90 | 0.08 | | | | |
| | West | 12.03 | 11.11 | 0.92 | 0.08 | | | | |
| | Northwest | 8.10 | 10.58 | 1.31 | 0.11 | | | | |
| | North | 6.08 | 6.35 | 1.04 | 0.09 | | | | |
| Elevation (m) (ASTER) | <47 | 43.64 | 22.92 | 0.53 | 0.11 | 0.00 | 0.50 | 0.50 | 3.57 |
| | 47–89 | 33.46 | 40.10 | 1.20 | 0.26 | | | | |
| | 89–156 | 14.33 | 33.33 | 2.33 | 0.50 | | | | |
| | 156–264 | 6.52 | 3.65 | 0.56 | 0.12 | | | | |
| | 264–577 | 2.06 | 0.00 | 0.00 | 0.00 | | | | |
| Elevation (m) (SRTM) | <60 | 52.93 | 26.56 | 0.50 | 0.11 | 0.00 | 0.41 | 0.41 | 3.05 |
| | 60–108 | 30.88 | 53.65 | 1.74 | 0.38 | | | | |
| | 108–178 | 9.16 | 17.19 | 1.88 | 0.41 | | | | |
| | 178–282 | 5.29 | 2.60 | 0.49 | 0.11 | | | | |
| | 282–577 | 1.75 | 0.00 | 0.00 | 0.00 | | | | |
| Elevation (m) (ALOS PALSAR) | <5 | 51.07 | 26.04 | 0.51 | 0.11 | 0.00 | 0.44 | 0.44 | 3.15 |
| | 5–54 | 32.46 | 52.08 | 1.60 | 0.35 | | | | |
| | 54–126 | 9.52 | 19.27 | 2.03 | 0.44 | | | | |
| | 126–229 | 5.22 | 2.60 | 0.50 | 0.11 | | | | |
| | 229–526 | 1.74 | 0.00 | 0.00 | 0.00 | | | | |

**Table A1.** *Cont.*

| Causal Factors | Classes | Area (%), $\frac{Aij}{Atotal}$ | Landslides (%), $\frac{Nij}{Ntotal}$ | FR, $\frac{Nij/Ntotal}{Aij/Atotal}$ | RF, $\frac{FRij}{\sum FRi}$ | MinRF | MaxRF | (MaxRF - MinRF) | PR |
|---|---|---|---|---|---|---|---|---|---|
| Elevation (m) (SOB) | <56 | 52.90 | 32.11 | 0.61 | 0.14 | 0.05 | 0.43 | 0.38 | 2.41 |
| | 56–111 | 24.38 | 45.79 | 1.88 | 0.43 | | | | |
| | 111–174 | 13.96 | 20.00 | 1.43 | 0.33 | | | | |
| | 174–248 | 4.91 | 1.05 | 0.21 | 0.05 | | | | |
| | 248–498 | 3.86 | 1.05 | 0.27 | 0.06 | | | | |
| Slope (°) (ASTER) | <4 | 33.20 | 12.50 | 0.38 | 0.06 | 0.06 | 0.31 | 0.31 | 2.22 |
| | 4–8 | 28.17 | 23.96 | 0.85 | 0.13 | | | | |
| | 8–14 | 21.88 | 30.73 | 1.40 | 0.22 | | | | |
| | 14–22 | 12.64 | 25.52 | 2.02 | 0.31 | | | | |
| | 22–52 | 4.10 | 7.29 | 1.78 | 0.28 | | | | |
| Slope (°) (SRTM) | <3 | 32.81 | 8.85 | 0.27 | 0.04 | 0.04 | 0.33 | 0.28 | 2.12 |
| | 3–8 | 30.18 | 26.56 | 0.88 | 0.15 | | | | |
| | 8–14 | 21.92 | 37.50 | 1.71 | 0.28 | | | | |
| | 14–22 | 11.59 | 22.92 | 1.98 | 0.33 | | | | |
| | 22–61 | 3.51 | 4.17 | 1.19 | 0.20 | | | | |
| Slope (°) (ALOS PALSAR) | <4 | 32.64 | 5.73 | 0.18 | 0.03 | 0.03 | 0.34 | 0.31 | 2.25 |
| | 4–9 | 28.63 | 27.60 | 0.96 | 0.16 | | | | |
| | 9–15 | 22.79 | 36.46 | 1.60 | 0.26 | | | | |
| | 15–23 | 12.29 | 25.52 | 2.08 | 0.34 | | | | |
| | 23–65 | 3.65 | 4.69 | 1.28 | 0.21 | | | | |
| Slope (°) (SOB) | <2 | 60.16 | 48.68 | 1.50 | 0.45 | 0.07 | 0.35 | 0.28 | 1.79 |
| | 2–4 | 22.75 | 28.04 | 0.81 | 0.16 | | | | |
| | 4–8 | 11.49 | 20.63 | 1.23 | 0.24 | | | | |
| | 8–14 | 4.50 | 1.59 | 1.80 | 0.35 | | | | |
| | 14–46 | 1.10 | 1.06 | 0.35 | 0.07 | | | | |
| TWI (ASTER) | <6 | 40.58 | 60.94 | 1.50 | 0.45 | 0.00 | 0.45 | 0.45 | 3.19 |
| | 6–8 | 29.20 | 29.17 | 1.00 | 0.30 | | | | |
| | 8–11 | 12.33 | 6.77 | 0.55 | 0.16 | | | | |
| | 11–14 | 10.90 | 3.12 | 0.29 | 0.09 | | | | |
| | >14 | 6.98 | 0.00 | 0.00 | 0.00 | | | | |
| TWI (SRTM) | <6 | 45.22 | 71.35 | 1.58 | 0.54 | 0.00 | 0.54 | 0.54 | 4.03 |
| | 6–8 | 26.17 | 23.96 | 0.92 | 0.31 | | | | |
| | 8–11 | 10.29 | 3.65 | 0.35 | 0.12 | | | | |
| | 11–14 | 12.04 | 1.04 | 0.09 | 0.03 | | | | |
| | >14 | 6.28 | 0.00 | 0.00 | 0.00 | | | | |
| TWI (ALOS PALSAR) | <6 | 44.46 | 76.04 | 1.71 | 0.60 | 0.00 | 0.60 | 0.60 | 4.30 |
| | 6–9 | 29.03 | 18.23 | 0.63 | 0.22 | | | | |
| | 9–12 | 9.11 | 3.65 | 0.40 | 0.14 | | | | |
| | 12–15 | 15.41 | 2.08 | 0.14 | 0.05 | | | | |
| | >15 | 1.99 | 0.00 | 0.00 | 0.00 | | | | |
| TWI (SOB) | <8 | 24.42 | 29.63 | 1.21 | 0.29 | 0.04 | 0.29 | 0.25 | 1.61 |
| | 8–11 | 32.60 | 36.51 | 1.12 | 0.27 | | | | |
| | 11–14 | 22.74 | 20.11 | 0.88 | 0.21 | | | | |
| | 14–17 | 17.03 | 13.23 | 0.78 | 0.19 | | | | |
| | 17–32 | 3.21 | 0.53 | 0.16 | 0.04 | | | | |

**Table A1.** *Cont.*

| Causal Factors | Classes | Area (%), $\frac{Aij}{Atotal}$ | Landslides (%), $\frac{Nij}{Ntotal}$ | FR, $\frac{Nij/Ntotal}{Aij/Atotal}$ | RF, $\frac{FRij}{\sum FRi}$ | MinRF | MaxRF | (MaxRF - MinRF) | PR |
|---|---|---|---|---|---|---|---|---|---|
| **SPI (ASTER)** | <−3 | 37.02 | 38.54 | 1.04 | 0.24 | 0.06 | 0.30 | 0.24 | 1.67 |
| | −3–0 | 9.55 | 2.60 | 0.27 | 0.06 | | | | |
| | 0–4 | 23.11 | 30.21 | 1.31 | 0.30 | | | | |
| | 4–6 | 23.56 | 23.44 | 0.99 | 0.23 | | | | |
| | >6 | 6.77 | 5.21 | 0.77 | 0.18 | | | | |
| **SPI (SRTM)** | <−7 | 32.38 | 38.54 | 1.19 | 0.27 | 0.05 | 0.27 | 0.21 | 1.67 |
| | −7–−3 | 11.23 | 2.60 | 0.23 | 0.05 | | | | |
| | −3–−1 | 29.73 | 30.21 | 1.02 | 0.23 | | | | |
| | −1–2 | 20.94 | 23.44 | 1.12 | 0.25 | | | | |
| | >2 | 5.72 | 5.21 | 0.91 | 0.20 | | | | |
| **SPI (ALOS PALSAR)** | <−5 | 21.13 | 17.19 | 0.81 | 0.27 | 0.05 | 0.27 | 0.21 | 1.55 |
| | −5–−1 | 21.11 | 15.63 | 0.74 | 0.05 | | | | |
| | −1–3 | 25.19 | 26.04 | 1.03 | 0.23 | | | | |
| | 3–5 | 25.87 | 34.90 | 1.35 | 0.25 | | | | |
| | >5 | 6.70 | 6.25 | 0.93 | 0.20 | | | | |
| **SPI (SOB)** | <−6 | 4.37 | 3.17 | 0.73 | 0.16 | 0.14 | 0.30 | 0.16 | 1.00 |
| | −6–−2 | 9.28 | 6.88 | 0.74 | 0.16 | | | | |
| | −2–1 | 27.88 | 18.52 | 0.66 | 0.14 | | | | |
| | 1–3.1 | 33.70 | 37.04 | 1.10 | 0.24 | | | | |
| | >3.1 | 24.77 | 34.39 | 1.39 | 0.30 | | | | |
| **Plan curvature (ASTER)** | Convex | 37.16 | 38.54 | 1.04 | 0.43 | 0.05 | 0.52 | 0.47 | 3.29 |
| | Flat | 15.76 | 2.08 | 0.13 | 0.05 | | | | |
| | Concave | 47.09 | 59.37 | 1.26 | 0.52 | | | | |
| **Plan curvature (SRTM)** | Convex | 37.79 | 45.31 | 1.20 | 0.50 | 0.01 | 0.50 | 0.49 | 3.64 |
| | Flat | 15.81 | 0.52 | 0.03 | 0.01 | | | | |
| | Concave | 46.41 | 54.17 | 1.17 | 0.49 | | | | |
| **Plan curvature (ALOS PALSAR)** | Convex | 37.38 | 50.00 | 1.34 | 0.54 | 0.00 | 0.60 | 0.60 | 4.30 |
| | Flat | 17.39 | 0.52 | 0.03 | 0.01 | | | | |
| | Concave | 45.22 | 49.48 | 1.09 | 0.44 | | | | |
| **Plan curvature (SOB)** | Concave | 33.42 | 41.05 | 1.23 | 0.43 | 0.20 | 0.43 | 0.23 | 1.44 |
| | Flat | 25.42 | 14.74 | 0.58 | 0.20 | | | | |
| | Complex | 41.15 | 44.21 | 1.07 | 0.37 | | | | |
| **Profile curvature (ASTER)** | Convex | 35.56 | 50.00 | 1.41 | 0.57 | 0.05 | 0.52 | 0.46 | 3.28 |
| | Flat | 12.81 | 1.56 | 0.12 | 0.05 | | | | |
| | Concave | 51.63 | 48.44 | 0.94 | 0.38 | | | | |
| **Profile curvature (SRTM)** | Convex | 36.20 | 54.69 | 1.51 | 0.62 | 0.00 | 0.62 | 0.62 | 4.68 |
| | Flat | 13.89 | 0.00 | 0.00 | 0.00 | | | | |
| | Concave | 49.91 | 45.31 | 0.91 | 0.38 | | | | |
| **Profile curvature (ALOS PALSAR)** | Convex | 36.80 | 54.69 | 1.49 | 0.61 | 0.00 | 0.61 | 0.61 | 4.43 |
| | Flat | 14.60 | 0.00 | 0.00 | 0.00 | | | | |
| | Concave | 48.60 | 45.31 | 0.93 | 0.39 | | | | |

**Table A1.** *Cont.*

| Causal Factors | Classes | Area (%), $\frac{Aij}{Atotal}$ | Landslides (%), $\frac{Nij}{Ntotal}$ | FR, $\frac{Nij/Ntotal}{Aij/Atotal}$ | RF, $\frac{FRij}{\sum FRi}$ | MinRF | MaxRF | (MaxRF - MinRF) | PR |
|---|---|---|---|---|---|---|---|---|---|
| **Profile curvature(SOB)** | Convex | 35.78 | 36.84 | 1.03 | 0.61 | | | | |
| | Flat | 23.17 | 12.11 | 0.52 | 0.00 | 0.00 | 0.61 | 0.61 | 3.92 |
| | Concave | 41.05 | 51.05 | 1.24 | 0.39 | | | | |
| **Distance from the drainage Networks (m)** | <1427 | 22.58 | 42.33 | 1.87 | 0.36 | 0.09 (A) | 0.36 (A) | 0.27 (A) | 1.91 (A) |
| | 1427–2853 | 25.63 | 22.75 | 0.89 | 0.17 | 0.09 (B) | 0.36 (B) | 0.27 (B) | 2.02 (B) |
| | 2853–4280 | 25.04 | 15.87 | 0.63 | 0.12 | 0.09 (C) | 0.36 (C) | 0.27 (C) | 1.94 (C) |
| | 4280–5885 | 19.48 | 8.99 | 0.46 | 0.09 | 0.09 (D) | 0.36 (D) | 0.27 (D) | 1.72 (D) |
| | >5885 | 7.26 | 10.05 | 1.38 | 0.26 | | | | |
| **Distance from the fault lines (m)** | <2358 | 24.62 | 30.16 | 1.23 | 0.30 | 0.00 (A) | 0.33 (A) | 0.33 (A) | 2.33 (A) |
| | 2358–4715 | 26.28 | 35.45 | 1.35 | 0.33 | 0.00 (B) | 0.33 (B) | 0.33 (B) | 2.46 (B) |
| | 4715–7191 | 23.73 | 27.51 | 1.16 | 0.28 | 0.00 (C) | 0.33 (C) | 0.33 (C) | 2.37 (C) |
| | 7191–10197 | 18.72 | 6.88 | 0.37 | 0.09 | 0.00 (D) | 0.33 (D) | 0.33 (D) | 2.09 (D) |
| | >10917 | 6.65 | 0.00 | 0.00 | 0.00 | | | | |
| **Rainfall (mm)** | <2446 | 28.68 | 7.94 | 0.28 | 0.05 | 0.05 (A) | 0.36 (A) | 0.31 (A) | 2.16 (A) |
| | 2446–2525 | 24.82 | 46.56 | 1.88 | 0.36 | 0.05 (B) | 0.36 (B) | 0.31 (B) | 2.29 (B) |
| | 2525–2606 | 21.43 | 23.81 | 1.11 | 0.21 | 0.05 (C) | 0.36 (C) | 0.31 (C) | 2.20 (C) |
| | 2606–2707 | 9.54 | 14.29 | 1.50 | 0.29 | 0.05 (D) | 0.36 (D) | 0.31 (D) | 1.94 (D) |
| | 2707–2864 | 15.53 | 7.41 | 0.48 | 0.09 | | | | |
| **Distance from the road networks (m)** | <1165 | 33.23 | 89.42 | 2.69 | 0.88 | 0.00 (A) | 0.88 (A) | 0.88 (A) | 6.26 (A) |
| | 1165–2854 | 32.02 | 8.99 | 0.28 | 0.09 | 0.00 (B) | 0.88 (B) | 0.88 (B) | 6.61 (B) |
| | 2854–4542 | 21.43 | 1.59 | 0.07 | 0.02 | 0.00 (C) | 0.88 (C) | 0.88 (C) | 6.38 (C) |
| | 4542–6552 | 8.54 | 0.00 | 0.00 | 0.00 | 0.00 (D) | 0.88 (D) | 0.88 (D) | 5.64 (D) |
| | 6552–10250 | 4.78 | 0.00 | 0.00 | 0.00 | | | | |
| **NDVI** | <0.1 | 0.06 | 0.00 | 0.00 | 0.00 | 0.02 (A) | 0.36 (A) | 0.34 (A) | 2.41 (A) |
| | 0.1–0.2 | 73.55 | 70.26 | 0.96 | 0.10 | 0.02 (B) | 0.36 (B) | 0.34 (B) | 2.54 (B) |
| | 0.2–0.3 | 18.40 | 0.00 | 0.00 | 0.00 | 0.02 (C) | 0.36 (C) | 0.34 (C) | 2.45 (C) |
| | 0.3–0.4 | 5.37 | 14.87 | 2.77 | 0.29 | 0.02 (D) | 0.36 (D) | 0.34 (D) | 2.17 (D) |
| | 0.4–0.5 | 2.62 | 14.87 | 5.67 | 0.60 | | | | |

**Table A1.** *Cont.*

| Causal Factors | Classes | Area (%), $\frac{Aij}{Atotal}$ | Landslides (%), $\frac{Nij}{Ntotal}$ | FR, $\frac{Nij/Ntotal}{Aij/Atotal}$ | RF, $\frac{FRij}{\sum FRi}$ | MinRF | MaxRF | (MaxRF - MinRF) | PR |
|---|---|---|---|---|---|---|---|---|---|
| Land use/ land cover | Vegetation | 73.55 | 70.26 | 0.96 | 0.10 | 0.00 (A) | 0.60 (A) | 0.60 (A) | 4.27 (A) |
| | Water bodies | 18.40 | 0.00 | 0.00 | 0.00 | 0.00 (B) | 0.60 (B) | 0.60 (B) | 4.52 (B) |
| | Bare land | 5.37 | 14.87 | 2.77 | 0.29 | 0.00 (C) | 0.60 (C) | 0.60 (C) | 4.36 (C) |
| | Built up | 2.62 | 14.87 | 5.67 | 0.60 | 0.00 (D) | 0.60 (D) | 0.60 (D) | 3.85 (D) |
| Land use /land cover change | Water-vegetation | 5.31 | 2.05 | 0.39 | 0.01 | 0.00 (A) | 0.23 (A) | 0.23 (A) | 1.60 (A) |
| | Water-bare Land | 0.70 | 0.51 | 0.73 | 0.02 | 0.00 (B) | 0.23 (B) | 0.23 (B) | 1.69 (B) |
| | Water-built up | 0.25 | 0.51 | 2.02 | 0.05 | 0.00 (C) | 0.23 (C) | 0.23 (C) | 1.63 (C) |
| | Vegetation-bare land | 3.54 | 12.82 | 3.62 | 0.09 | 0.00 (D) | 0.23 (D) | 0.23 (D) | 1.44 (D) |
| | Vegetation-built up | 1.74 | 9.23 | 5.31 | 0.14 | | | | |
| | Built up-vegetation | 0.81 | 1.54 | 1.91 | 0.05 | | | | |
| | Built up-bare land | 0.23 | 0.51 | 2.20 | 0.06 | | | | |
| | Bare land-vegetation | 2.09 | 7.18 | 8.65 | 0.23 | | | | |
| | Bare land-built up | 0.45 | 3.59 | 3.44 | 0.09 | | | | |
| | No change | 84.88 | 62.06 | 0.73 | 0.02 | | | | |
| Geology | Dihing and Dupi Tila formation | 4.65 | 19.07 | 4.10 | 0.40 | 0.00 (A) | 0.40 (A) | 0.40 (A) | 2.83 (A) |
| | Boka Bil formation | 28.92 | 30.41 | 1.05 | 0.10 | 0.00 (B) | 0.40 (B) | 0.40 (B) | 3.00 (B) |
| | Bhuban formation | 8.97 | 16.49 | 1.84 | 0.18 | 0.00 (C) | 0.40 (C) | 0.40 (C) | 2.89 (C) |
| | Tipam sandstone | 12.41 | 27.84 | 2.24 | 0.22 | 0.00 (D) | 0.40 (D) | 0.40 (D) | 2.56 (D) |
| | Valley alluvium and colluvium | 0.46 | 0.00 | 0.00 | 0.00 | | | | |
| | Dupi tile formation | 14.73 | 3.61 | 0.25 | 0.02 | | | | |
| | Water bodies | 26.46 | 0.00 | 0.00 | 0.00 | | | | |
| | Girujan clay | 3.41 | 2.58 | 0.76 | 0.07 | | | | |

## Appendix E

**Table A2.** Coefficients of significant causal factors of DEM based: LR_ASTER_DEM; LR_SRTM_DEM; LR_ALOS_DEM and LR_SOB_DEM and all factors based: LR_ASTER; LR_SRTM; LR_ALOS and LR_SOB models.

| Factors Used | DEM | Causal Factors | ß | Standard Error | Wald | Sig | Exp(ß) |
|---|---|---|---|---|---|---|---|
| **DEM based** | ASTER | Elevation | 0.06 | 0.01 | 29.72 | 0.00 | 1.06 |
| | | Slope | 0.26 | 0.03 | 78.02 | 0.00 | 1.30 |
| | | SPI | 0.08 | 0.04 | 5.28 | 0.02 | 1.09 |
| | | Constant | −6.99 | 1.06 | 43.10 | 0.00 | 0.00 |
| | SRTM | Slope | 0.14 | 0.02 | 44.32 | 0.00 | 1.15 |
| | | Elevation | 0.04 | 0.01 | 15.93 | 0.00 | 1.04 |
| | | TWI | 0.04 | 0.01 | 12.09 | 0.00 | 1.04 |
| | | Constant | −4.70 | 0.50 | 86.89 | 0.00 | 0.01 |
| | ALOS PALSAR | Aspect | 0.09 | 0.05 | 3.818 | 0.04 | 1.09 |
| | | TWI | 0.02 | 0.01 | 5.94 | 0.02 | 1.02 |
| | | Slope | 0.05 | 0.02 | 23.18 | 0.00 | 1.05 |
| | | Elevation | 0.05 | 0.01 | 18.17 | 0.00 | 1.05 |
| | | Plan | 0.05 | 0.03 | 3.33 | 0.07 | 1.05 |
| | | Constant | −6.77 | 1.40 | 23.35 | 0.00 | 0.00 |
| | SOB | Elevation | 0.04 | 0.01 | 26.72 | 0.00 | 1.04 |
| | | Aspect | 0.05 | 0.01 | 23.21 | 0.00 | 1.05 |
| | | Constant | −1.68 | 0.25 | 47.05 | 0.00 | 0.19 |
| **All factors** | ASTER | Elevation | 0.08 | 0.02 | 26.61 | 0.00 | 1.087 |
| | | Slope | 0.31 | 0.04 | 58.81 | 0.00 | 1.37 |
| | | Change [1] | 0.18 | 0.05 | 15.72 | 0.00 | 1.20 |
| | | Drainage [2] | 0.07 | 0.02 | 11.96 | 0.00 | 1.07 |
| | | Rainfall | −0.04 | 0.02 | 4.24 | 0.04 | 0.96 |
| | | Road [3] | 0.04 | 0.01 | 37.47 | 0.00 | 1.04 |
| | | Constant | −9.58 | 1.14 | 71.27 | 0.00 | 0.00 |
| | SRTM | Slope | 0.16 | 0.03 | 42.52 | 0.00 | 1.175 |
| | | Elevation | 040 | 0.01 | 8.17 | 0.00 | 1.041 |
| | | SPI | 0.24 | 0.09 | 7.34 | 0.01 | 1.266 |
| | | Land [4] | 0.08 | 0.02 | 20.74 | 0.000 | 1.080 |
| | | Drainage [2] | 0.06 | 0.02 | 11.69 | 0.001 | 1.063 |
| | | Rainfall | −0.04 | 0.02 | 4.94 | 0.026 | 0.961 |
| | | Road [3] | 0.03 | 0.01 | 27.71 | 0.000 | 1.025 |
| | | Fault [5] | 0.08 | 0.03 | 9.82 | 0.002 | 1.082 |
| | | Constant | −14.59 | 2.58 | 31.91 | 0.000 | 0.000 |
| | ALOS PALSAR | Aspect | 0.12 | 0.051 | 5.464 | 0.02 | 1.126 |
| | | Slope | 0.13 | 0.02 | 42.01 | 0.00 | 1.14 |
| | | Elevation | 0.06 | 0.01 | 16.97 | 0.00 | 1.06 |
| | | Change [1] | 0.14 | 0.04 | 11.65 | 0.00 | 1.15 |
| | | Drainage [2] | 0.06 | 0.02 | 13.48 | 0.00 | 1.06 |
| | | Road [3] | 0.03 | 0.01 | 32.67 | 0.00 | 1.02 |
| | | Constant | −8.36 | 1.00 | 69.71 | 0.00 | 0.00 |
| | SOB | Aspect | 0.05 | 0.01 | 12.13 | 0.00 | 1.05 |
| | | Land [4] | 0.05 | 0.01 | 16.62 | 0.00 | 1.05 |
| | | Fault [5] | 0.07 | 0.02 | 14.15 | 0.00 | 1.07 |
| | | Road [3] | 0.03 | 0.04 | 80.38 | 0.00 | 1.04 |
| | | Constant | −5.05 | 0.64 | 61.70 | 0.00 | 0.01 |

1 = Land use/land cover change; 2 = distance from the drainage networks; 3 = distance from the road networks; 4 = land use/land cover; 5 = distance from the fault lines.

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
