# Peer review of "Evaluating the Effects of Digital Elevation Models in Landslide Susceptibility Mapping in Rangamati District, Bangladesh"

_remotesensing, doi:10.3390/rs12172718_

Round 1
Reviewer 1 Report
Overview
The authors demonstrate quantitative differences in landslide susceptibility maps based on DEMs (four), assessments (three), and inputs (two). Results generally show global DEMs perform better than the 25m Survey of Bangladesh DEM.
General Comments
Moderate English changes required. Please make sure to edit paper for grammar and wording. For example, suggested edits (in bold) for the first three sentences of the abstract: "Digital Elevation Models (DEMs) is are the most obvious factors in landslide susceptibility assessment. Many landslide casual factors are often generated from DEMs. Most of the studies on landslide susceptibility assessment often relies rely on DEMs that are available at free of charge."
The final paragraph of Section 1 (lines 96-116) needs editing.
The authors claim "a few attempts have been made to compare the performance of DEMs with different spatial resolutions in landslide susceptibility assessment" (lines 82-84). Five studies are provided as evidence. The authors then state "[t]hese past studies further indicated that the performance of DEM [sic] is context-dependent meaning the performance of a DEM in a region may not be assumed to be similar in another region" (lines 96-97) and "[t]herefore, it is an upmost [sic, it should say utmost] need to have comparative assessments of DEMs in various contexts. With this vain, this study contextualized landslide susceptibility in Bangladesh and compared the performance of different DEMs and modeling techniques" (lines 99-101). Is this the first comparison in Bangladesh? If so, maybe make mention of this fact instead of stating that DEMs have not been compared as much as the authors deem necessary. I have found other papers that compare DEMs for landslide purposes, including Impact of DEM-derived factors and analytical hierarchy process on landslide susceptibility mapping in the region of Roznow Lake, Poland by Pawluszek & Borkowski (2017) and Evolution of a diachronic landslide by comparison between different DEMs obtained from Digital Photogrammetry Techniques in Las Alpujarras (Granada, Southern Spain) by Fernandez et al. (2011), for example.
Study area paragraph (lines 118-128) needs editing.
Figure 1. The 200 km scale bars in Figure 1a and (presumably) 1b cannot both be correct. It looks like the three Upazilas are over 200 km wide in the upper part of Figure 1, yet the entirety of Rangamati District is much less than 200 km in the bottom left image (when viewing Bangladesh as a country).
Figure 1. What are the non-landslide locations and landslides locations? Is this figure referenced in the text?
Section 3.2. This comment is about header organization. Sections 3.2.6 Rainfall, 3.2.7 Distance-Based Causal Factors, 3.2.8 Normalized Difference Vegetation Index (NDVI), 3.2.9 Geology, and 3.2.10 Land use land cover should all be renumbered. Right now they are nested within 3.2 Landslide Causal Factors, but rainfall, distance-based causal factors, NDVI, geology, and land use/land cover are not landslide causal factors. Instead, they should be renumbered as follows:
3.3 Rainfall
3.4 Distance-based Causal Factors
3.5 Normalized Difference Vegetation Index (NDVI)
3.6 Geology
3.7 Land Use/Land Cover
Table 1. This paper is acronym heavy. According to Table 1, many acronyms are repeated. This is utterly confusing. Please take a look at the final column and fix the acronyms. As an example, MFR_SOB is used three times: (1) Modified Frequency Ratio + All 15 factors + SOB DEM, (2) Logistic Regression + All 15 factors + SOB DEM, and (3) Random Forest + All 15 factors + SOB DEM.
Another comment about acronyms. There are too many and they are used way too often. For example, I can barely read lines 383-400 without pausing and looking back at what each acronym means. To properly read this paper, I need a cheat sheet with all the acronyms to reference.
Landslide Susceptibility Index (LSI). Can the authors provide an interpretation of LSI values? As a reader, I am not sure which LSI values are low or average or high, which values indicate immediate danger, etc.? When the authors say something like "The LSI of MFR_ASTER DEM ranged from 1613.00 to 20370.10. The LSIs for MFR_SRTM, MFR_ALOS, and SOB DEMs ragned from 1314.40 to 22300.34 and 1234.95 to 22180.24 and 1995.7 to 17316.9, respectively" (lines 391-394), it means absolutely nothing to me. I need the authors to provide me with their interpretation of what these values mean for me to glean any understanding of the significance of these numbers. The authors don't even provide units or a scale (see equation 5 in lines 284-285). What do these numbers mean? ...Then in lines 401-402 the authors state "We used the Jenks natural break method to classify the LSIs into five susceptibility zones: very low, low, moderate, high, and very high." That's wonderful, but please provide the numeric values of these breaks. Why provide us with a lot of LSI values and then never tell us how they are divided?
Jenks Natural Break. Are all variables mathematically classified used Jenks Natural Break? Are there any variables where classification was decided using another approach (e.g., is there any real-world meaning behind these classifications, or are they purely mathematical)? Using a 100% mathematical approach can give the semblance of a quantitative approach, but sometimes a purely mathematical approach does not mean anything and can be equally arbitrary.
Figures 2 and 3. These figures are set up so the reader compares the method for each DEM. Each row provides a DEM (row 1 = ASTER, row 2 = SRTM, etc.). As I read the manuscript, my eyes naturally scroll from left to right along the rows. If the authors wish the reader to compare MFR_ASTER DEM (7 variables) to MFR_ASTER DEM (15 variables), it is very difficult to do so in the current format. The text is written in such a way that requires the reader to flip back and forth between Figure 2 and Figure 3. Maybe the authors should reconsider how these figures are formatted. To drive my point home, here is the order figures are referenced between lines 401 and 488: Fig 2d, Fig 2a-c, Fig 3a-d, Fig 2h, Fig 2f, Fig 2g, Fig 3h, Fig 3e-g, Fig 3e, Fig 3f-g, Fig 4a, Fig 4b, Fig 4b (again), Fig 2l, Fig 2i-k, Fig 2i, Fig 4a, and Fig 3i-l. Please reconsider how these figures are formatted. The authors seem to be making the following comparisons: difference between Fig 2a-d and Fig 3a-d, difference between Fig 2e-h and Fig 3e-h, and difference between Fig 2i-l and Fig 3i-l. Remaking these figures so that these comparisons are side-by-side would be quite helpful to the reader and follow the text better. Right now, to be honest, the text and figures are a complicated web requiring too many page turns -- as a reader, I start skimming the text because it is (1) too much work for the information provided, and (2) combined with the acronyms, there is just too much to keep track of that I find myself losing interest in the content of this paper. Basically, and again to be honest, this paper is showing something very simple (yes, using different DEMs will yield different landslide susceptibility maps) in the most complex way possible.
Figure 4. Please fix the letters "a" and "b" in the figure. They are misaligned with each other. Additionally, why use "AP" as an acronym of ALOS PALSAR in the graphs? It looks like there is plenty of space to spell out ALOS PALSAR. This paper does not need superfluous acronyms.
Figure 4. Which method is being shown in Figures 4a and 4b? The text (lines 465-473) indicate the Random Forest method is shown in Figure 4, but the figure caption makes no mention of method used.
Figure 5 and Section 5.4.1.1. I present here another issue with how a figure is set up and how the information is presented in the text. Between lines 500 and 529, the authors compare the following: Figures 5a and 5d, Figures 5b and 5e, and Figures 5c and 5f are not mentioned. Why not change the lettering so that you compare Figures 5a and 5b, 5c and 5d, and 5e and 5f? This allows the figures to be listed in order in the text, and the reader does not have to think as much about which letters are being compared.
Figure 6 and Section 5.4.1.2. Same comment about Figure 6 as I have about Figure 5 above. The figure lettering is more complicated than it needs to be. As a reader, by this point in the paper, I am very tired of flipping back and forth between figures for minimal gain. It could be simpler.
Figures 5 and 6. Again, do we need the AP acronym for ALOS PALSAR?
Figures 5 and 6. Please increase figure quality. Text is grainy (low resolution?).
"The AUCs of success rate curves (Fig 5a) showed that MFR_SRTM_DEM falls under the good category while MFR_ASTER_DEM and MFR_ALOS_DEM fall under fare [sic] category. But MFR_SOB_DEM falls under the poor category." (lines 502-504). Are the good, fair, and poor categories ever numerically defined? There is a "fail category" (line 521)?
Section 5.4.1.2. Can the authors provide an explanation (or educated guess) as to why accuracy fell with the MFR model but increased with the LR and RF models?
Conclusion. The authors state "...the performances of ALOS PASAR and SRTM were comparable" (lines 624-625). Yet earlier in the same paragraph state "Spatial convergence analysis showed that the Global DEMs (e.g. ASTER, SRTM, ALOS PALSAR) based susceptibility maps have similar spatial appearances and classified 40-45% of the study area into the same susceptibility zones." (lines 617-619). "Similar spatial appearances" is qualitative. I disagree that ALOS PALSAR and SRTM "were comparable" if they only spatially agree 40-45% of the time. That seems to be pretty weak in my opinion. It would be interesting to note the classification differences, e.g., does ALOS PALSAR show low susceptibility while SRTM shows high susceptibility at the same location, or are the differences minimal (high susceptibility to very high susceptibility)? Because I can rephrase the author's sentence to say 'ALOS PALSAR and SRTM were not comparable and spatially differed over 55-60% of the study area based on susceptibility classification zones.' When phrased like that, it sounds less impressive. Although this paper is exceptionally long, I feel this is an important issue to examine closer -- just how different these susceptibility maps are, because if the difference is large (low susceptibility to high susceptibility) it would change the practical use of these maps greatly (the high variability does not promote confidence for those who live in the impacted areas).
Line-by-Line Comments
Lines 75 and 79: Change "...remote sensing sensors" to "...remote sensors."
Line 97: Add punctuation at end of sentence
Line 124: The authors reference Figure 3a here. I believe the text should reference Figure 1a instead. Additionally, the reference to Figure 1 (or Figure 3) does not make sense with the sentence, which lists the geology of this area -- neither figure shows the geology of this area.
Lines 145-146: The authors say "seven factors were derived" but only list six: (1) elevation, (2) slope plan, (3) profile curvatures, (4) TWI, (5) SPI, and (6) aspect. Based on the section headers (3.2.1-3.2.5), there should be slope angle, slope plan, and slope curvature as factors. Please clarify.
Lines 188 and 258: Two different equations are labeled as "Equation 2"
Line 192: Change "control" to "controls"
Lines 273-275: Please fix grammar of this sentence, particularly "...the higher is the PR value the stronger is the association..." sounds awkward.
Lines 284 and 299: Two different equations are labeled as "Equation 5"
Lines 312-313: "High predictive performance"? RF can predict landslide location, but not timing, surely?
Lines 406-408: Reword this sentence so that Fig 2a-c is mentioned before Figure 2d. This keeps chronology intact.
Line 447: Change "LR_ASETR" to "LR_ASTER"
Lines 505, 506, and 523: Change "fare" to "fair"
Line 549: Is a word missing at the end of this line?
Line 566: Remove parentheses around the number two in "Table (2)"
Line 568: Maybe change "...between the models" to "...between the DEMs"?
Author Response
Reviewer #1:
Response
Thank you for the comments and suggestions. These comments and suggestions have enabled us to improve the quality of our manuscript. We have accepted most of the corrections. Please check below the answers for each of the concerns about the manuscript.
- Moderate English changes required. Please make sure to edit paper for grammar and wording. For example, suggested edits (in bold) for the first three sentences of the abstract: "Digital Elevation Models(DEMs) isare the most obvious factors in landslide susceptibility assessment. Many landslide casual factors are often generated from DEMs. Most of the studies on landslide susceptibility assessment often relies rely on DEMs that are available at free of charge."
Response: Thank you for pointing out this problem. We have gone through the whole paper again and made necessary corrections.
- The final paragraph of Section 1 (lines 96-116) needs editing.
The authors claim "a few attempts have been made to compare the performance of DEMs with different spatial resolutions in landslide susceptibility assessment" (lines 82-84). Five studies are provided as evidence. The authors then state "[t]hese past studies further indicated that the performance of DEM [sic] is context-dependent meaning the performance of a DEM in a region may not be assumed to be similar in another region" (lines 96-97) and "[t]herefore, it is an upmost [sic, it should say utmost] need to have comparative assessments of DEMs in various contexts. With this vain, this study contextualized landslide susceptibility in Bangladesh and compared the performance of different DEMs and modeling techniques" (lines 99-101). Is this the first comparison in Bangladesh? If so, maybe make mention of this fact instead of stating that DEMs have not been compared as much as the authors deem necessary. I have found other papers that compare DEMs for landslide purposes, including Impact of DEM-derived factors and analytical hierarchy process on landslide susceptibility mapping in the region of Roznow Lake, Poland by Pawluszek & Borkowski (2017) and Evolution of a diachronic landslide by comparison between different DEMs obtained from Digital Photogrammetry Techniques in Las Alpujarras (Granada, Southern Spain) by Fernandez et al. (2011), for example.
Response: Thank you for pointing out these issues. Yes, this is the first comparison in Bangladesh. We have revised these lines. Please check lines 90-100
- Study area paragraph (lines 118-128) needs editing.
Response: Thank you for pointing out these issues. We have done the necessary edits.
- Figure 1. What are the non-landslide locations and landslides locations? Is this figure referenced in the text?
Response: To train the models: logistic regression and random forest; we need presence and absence data. Presence data means landslide locations. These locations are the locations where landslides previously occurred. Non-Landslide locations are pseudo-absence data. We selected same number (same as landslide locations) of pseudo non-landslide locations aka absence data randomly. We have added the term in the caption of Figure 1.
- Section 3.2. This comment is about header organization. Sections 3.2.6 Rainfall, 3.2.7 Distance-Based Causal Factors, 3.2.8 Normalized Difference Vegetation Index (NDVI), 3.2.9 Geology, and 3.2.10 Land use land cover should all be renumbered. Right now they are nested within 3.2 Landslide Causal Factors, but rainfall, distance-based causal factors, NDVI, geology, and land use/land cover are not landslide causal factors. Instead, they should be renumbered as follows:
3.3 Rainfall
3.4 Distance-based Causal Factors
3.5 Normalized Difference Vegetation Index (NDVI)
3.6 Geology
3.7 Land Use/Land Cover
Response: Thank you for the suggestion. We have accepted the suggestion.
- Table 1. This paper is acronym heavy. According to Table 1, many acronyms are repeated. This is utterly confusing. Please take a look at the final column and fix the acronyms. As an example, MFR_SOB is used three times: (1) Modified Frequency Ratio + All 15 factors + SOB DEM, (2) Logistic Regression + All 15 factors + SOB DEM, and (3) Random Forest + All 15 factors + SOB DEM.
Another comment about acronyms. There are too many and they are used way too often. For example, I can barely read lines 383-400 without pausing and looking back at what each acronym means. To properly read this paper, I need a cheat sheet with all the acronyms to reference.
Response: Thank you for raising these issues. We have revised the paper to remove unnecessary acronyms.
- Landslide Susceptibility Index (LSI). Can the authors provide an interpretation of LSI values? As a reader, I am not sure which LSI values are low or average or high, which values indicate immediate danger, etc.? When the authors say something like "The LSI of MFR_ASTER DEM ranged from 1613.00 to 20370.10. The LSIs for MFR_SRTM, MFR_ALOS, and SOB DEMs ragned from 1314.40 to 22300.34 and 1234.95 to 22180.24 and 1995.7 to 17316.9, respectively" (lines 391-394), it means absolutely nothing to me. I need the authors to provide me with their interpretation of what these values mean for me to glean any understanding of the significance of these numbers. The authors don't even provide units or a scale (see equation 5 in lines 284-285). What do these numbers mean? ...Then in lines 401-402 the authors state "We used the Jenks natural break method to classify the LSIs into five susceptibility zones: very low, low, moderate, high, and very high." That's wonderful, but please provide the numeric values of these breaks. Why provide us with a lot of LSI values and then never tell us how they are divided?
Response: Thank you for pointing out this important issue. We have given a shot description. Please check lines 394-397. We have provided the numeric values on the maps of Figure 2 and 3.
- Jenks Natural Break. Are all variables mathematically classified used Jenks Natural Break? Are there any variables where classification was decided using another approach (e.g., is there any real-world meaning behind these classifications, or are they purely mathematical)? Using a 100% mathematical approach can give the semblance of a quantitative approach, but sometimes a purely mathematical approach does not mean anything and can be equally arbitrary.
Response: We have tested various approach available in ArcGIS 10.7 for classification. We have found natural break methods gives the best result. Therefore, we used natural break method in our study.
- Figures 2 and 3. These figures are set up so the reader compares the method for each DEM. Each row provides a DEM (row 1 = ASTER, row 2 = SRTM, etc.). As I read the manuscript, my eyes naturally scroll from left to right along the rows. If the authors wish the reader to compare MFR_ASTER DEM (7 variables) to MFR_ASTER DEM (15 variables), it is very difficult to do so in the current format. The text is written in such a way that requires the reader to flip back and forth between Figure 2 and Figure 3. Maybe the authors should reconsider how these figures are formatted. To drive my point home, here is the order figures are referenced between lines 401 and 488: Fig 2d, Fig 2a-c, Fig 3a-d, Fig 2h, Fig 2f, Fig 2g, Fig 3h, Fig 3e-g, Fig 3e, Fig 3f-g, Fig 4a, Fig 4b, Fig 4b (again), Fig 2l, Fig 2i-k, Fig 2i, Fig 4a, and Fig 3i-l. Please reconsider how these figures are formatted. The authors seem to be making the following comparisons: difference between Fig 2a-d and Fig 3a-d, difference between Fig 2e-h and Fig 3e-h, and difference between Fig 2i-l and Fig 3i-l. Remaking these figures so that these comparisons are side-by-side would be quite helpful to the reader and follow the text better. Right now, to be honest, the text and figures are a complicated web requiring too many page turns -- as a reader, I start skimming the text because it is (1) too much work for the information provided, and (2) combined with the acronyms, there is just too much to keep track of that I find myself losing interest in the content of this paper. Basically, and again to be honest, this paper is showing something very simple (yes, using different DEMs will yield different landslide susceptibility maps) in the most complex way possible.
Response: Thank you for pointing out this problem. We have rearranged the order of the maps. Please check Figure 2 and 3.
- Figure 4. Please fix the letters "a" and "b" in the figure. They are misaligned with each other. Additionally, why use "AP" as an acronym of ALOS PALSAR in the graphs? It looks like there is plenty of space to spell out ALOS PALSAR. This paper does not need superfluous acronyms.
Figure 4. Which method is being shown in Figures 4a and 4b? The text (lines 465-473) indicate the Random Forest method is shown in Figure 4, but the figure caption makes no mention of method used.
Response: We have revised the caption of Figure 4. Please check lune 505.
- Figure 5 and Section 5.4.1.1. I present here another issue with how a figure is set up and how the information is presented in the text. Between lines 500 and 529, the authors compare the following: Figures 5a and 5d, Figures 5b and 5e, and Figures 5c and 5f are not mentioned. Why not change the lettering so that you compare Figures 5a and 5b, 5c and 5d, and 5e and 5f? This allows the figures to be listed in order in the text, and the reader does not have to think as much about which letters are being compared.
Response: Thank you for raising this issue. We have reordered the figure numbers.
- Figure 6 and Section 5.4.1.2. Same comment about Figure 6 as I have about Figure 5 above. The figure lettering is more complicated than it needs to be. As a reader, by this point in the paper, I am very tired of flipping back and forth between figures for minimal gain. It could be simpler.
Figures 5 and 6. Again, do we need the AP acronym for ALOS PALSAR?
Figures 5 and 6. Please increase figure quality. Text is grainy (low resolution?).
Response: We have reordered the figures. We have given the full form of ALOS PALSAR. Previously we have 600dpi image and to increase the quality we have provided 1200 dpi.
- "The AUCs of success rate curves (Fig 5a) showed that MFR_SRTM_DEM falls under the good category while MFR_ASTER_DEM and MFR_ALOS_DEM fall under fare [sic] category. But MFR_SOB_DEM falls under the poor category." (lines 502-504). Are the good, fair, and poor categories ever numerically defined? There is a "fail category" (line 521)?
Response: Yes they are numerically defined. This category is provided by Rasyid et al.. AUC value ranges from 0-1 or 0-100% and it can be grouped into the following categories: 0.50-0.60 (Fail); 0.60-0.70 (Poor); 0.70-0.80 (Fair); 0.80-0.90 (Good), and 0.90-1.00 (Excellent) [53].
- Section 5.4.1.2. Can the authors provide an explanation (or educated guess) as to why accuracy fell with the MFR model but increased with the LR and RF models?
Response: We have discussed it in the newly added discussion section. Please check lines 628-644.
- Conclusion. The authors state "...the performances of ALOS PASAR and SRTM were comparable" (lines 624-625). Yet earlier in the same paragraph state "Spatial convergence analysis showed that the Global DEMs (e.g. ASTER, SRTM, ALOS PALSAR) based susceptibility maps have similar spatial appearances and classified 40-45% of the study area into the same susceptibility zones." (lines 617-619). "Similar spatial appearances" is qualitative. I disagree that ALOS PALSAR and SRTM "were comparable" if they only spatially agree 40-45% of the time. That seems to be pretty weak in my opinion. It would be interesting to note the classification differences, e.g., does ALOS PALSAR show low susceptibility while SRTM shows high susceptibility at the same location, or are the differences minimal (high susceptibility to very high susceptibility)? Because I can rephrase the author's sentence to say 'ALOS PALSAR and SRTM were not comparable and spatially differed over 55-60% of the study area based on susceptibility classification zones.' When phrased like that, it sounds less impressive. Although this paper is exceptionally long, I feel this is an important issue to examine closer -- just how different these susceptibility maps are, because if the difference is large (low susceptibility to high susceptibility) it would change the practical use of these maps greatly (the high variability does not promote confidence for those who live in the impacted areas).
Response: Thank you for raising this issue and suggestions. We have deleted these lines and rewritten the conclusion part.
- Line-by-Line Comments
Lines 75 and 79: Change "...remote sensing sensors" to "...remote sensors."
Line 97: Add punctuation at end of sentence
Line 124: The authors reference Figure 3a here. I believe the text should reference Figure 1a instead. Additionally, the reference to Figure 1 (or Figure 3) does not make sense with the sentence, which lists the geology of this area -- neither figure shows the geology of this area.
Lines 145-146: The authors say "seven factors were derived" but only list six: (1) elevation, (2) slope plan, (3) profile curvatures, (4) TWI, (5) SPI, and (6) aspect. Based on the section headers (3.2.1-3.2.5), there should be slope angle, slope plan, and slope curvature as factors. Please clarify.
Lines 188 and 258: Two different equations are labeled as "Equation 2"
Line 192: Change "control" to "controls"
Lines 273-275: Please fix grammar of this sentence, particularly "...the higher is the PR value the stronger is the association..." sounds awkward.
Lines 284 and 299: Two different equations are labeled as "Equation 5"
Lines 312-313: "High predictive performance"? RF can predict landslide location, but not timing, surely?
Lines 406-408: Reword this sentence so that Fig 2a-c is mentioned before Figure 2d. This keeps chronology intact.
Line 447: Change "LR_ASETR" to "LR_ASTER"
Lines 505, 506, and 523: Change "fare" to "fair"
Line 549: Is a word missing at the end of this line?
Line 566: Remove parentheses around the number two in "Table (2)"
Line 568: Maybe change "...between the models" to "...between the DEMs"?
Response: We have accepted the corrections.
Reviewer 2 Report
Please see attachment.

Author Response
Reviewer #2:
Response
Thank you for the comments and suggestions. These comments and suggestions have enabled us to improve the quality of our manuscript. We have accepted most of the corrections. Please check below the answers for each of the concerns about the manuscript.
- Although the manuscript topic is interesting, the real research application of the findings and conclusions must be better explained in detail. I suggest adding a paragraph(s) to emphasize more the real applicability of the research finding in the future. Please provide more state-of-the-art references in the field. Proposed papers that can help authors to improve manuscript in the field of DEM evaluation: “Evaluating digital elevation models for glaciologic applications: An example from Nevado Coropuna, Peruvian Andes”; “A novel automated method for the improvement of photogrammetric DTM accuracy in forests”, etc.
Response: Thank you for your advice. We have added the discussion section. We have read the paper and cited in the article. We have discussed the application of the findings in discussion and conclusion section. We think, this per will help researchers in Bangladesh to chose DEM in landslide susceptibility assessment.
- Please introduce abbreviations if you want to use it in the abstract or in the manuscript text. Please introduce the abbreviation upon the first appearance in the manuscript text. This also applies to the abbreviations introduced in the abstract.
Response: We have revised the manuscript to address this issue.
- Please, correct technical problems with equations.
Response: Thank you for pointing out this serious issue. We have re-numbered the equitation and revised them.
- The variable names must have the same font style and size in equations, on figures, tables, and in the manuscript text. Please describe/introduce all variables used in equations or on figures in the manuscript text
Response: Thank you for raising these issues. We have revised the manuscript to address these issues.
- Please, double-check all references and reference style.
Response: We have revised the reference list.
Reviewer 3 Report
- The originality, accuracy and completeness of the work are satisfactory.
- The arrangements of references should be consistent throughout the references list.
- The authors are suggested to revise some typing errors in the manuscript.

Author Response
Reviewer #3:
Response
Thank you for the comments and suggestions. These comments and suggestions have enabled us to improve the quality of our manuscript. We have accepted most of the corrections. Please check below the answers for each of the concerns about the manuscript.
- The originality, accuracy, and completeness of the work are satisfactory’.
Response: Thank you for your comment.
- The arrangement of reference should be consistent throughout the reference list
Response: Thank you for pointing out this issue. We have revised the reference list.
- The authors are suggested to revise some typing errors in the manuscript.
Response: We have revised them
Reviewer 4 Report
This study could provide valuable guidance for conducting landslide susceptibility assessment in Bangladesh and similarly situated regions world-wide. However, the presentation of the results makes it difficult for the reader to understand what the key findings of this study are.
Currently, the Results section is very long and difficult to read because of the numerous acronyms, including the "model_data_DEM" acronyms. I understand the authors use these acronyms to simplify the description of a complicated cross-tabulation, but I think the Results section could be shortened by synthesizing the results across different data sources (e.g., SRTM, ASTER, etc.) and across different DEM sources. I think the authors have provided the reader with much more detail than is necessary. This hampers the reader's ability to understand what the key points of this study are. In particular, while the study reads as a model evaluation (i.e., which modeling approach provides the best results?), very little is said about how the different models (MFR, logistic regression, and random forest) compare to one another; instead, the authors focus on what combination of data inputs work best for each model. I should think they would want to make an overall recommendation about which model works best, as well. For example, they write in the Conclusion: "For the MFR model, we recommend complimenting the SOB DEM derived causal factors with other common causal factors (e.g. geology, land use/land cover)" (Lines 614-616). However, the study seems to indicate that the SOB DEM is generally unreliable ("Therefore, we argue that causal factors derived from the SOB DEM cannot explain the probability of landslides in the study area" Lines 613-614). Rather than recommending a specific approach that maximizes the utility of the SOB DEM, shouldn't the recommendation be not to use the SOB DEM in the first place? The authors do say this in the end (Line 623: "we suggest extreme caution before using SOB DEM for hilly areas in Bangladesh"). However, the authors could save space (and readers' time) by getting to the point much sooner, rather than describing the nuance of every possible model-data comparison. This is just one example of several issues throughout the paper where excessive description (and appendices) could be eliminated to make the paper more clear to readers.
There are also a couple of substantial technical issues with the paper in its current form. First, I think it is a poor choice to classify the values of the causal factors used in logistic regression and random forest. While this classification is necessary for the frequency ratio approach, it only serves to artificially restrict the variance in the predictive features for logistic regression and random forest. I have seen others do this before in model evaluation studies, with the idea that it facilitates a "fair" comparison between models that require discrete inputs (e.g., frequency ratio) and those that don't (e.g., logistic regression, random forest). However, this is a dubious argument, because forcing models that can accept continuous data to use arbitrarily classified inputs is putting a severe handicap on the more flexible models. Conversely, those models that require discrete data ought to be evaluated as to whether or not they provide higher predictive accuracy given their strong assumptions.
Second, the impact of multicollinearity on variable importance is not explored here. While it would not affect the predictive accuracy of the logistic regression and random forest models, it can substantially impact variable importance rankings or significance test. If the authors wish to make conclusions about which predictive features or "causal factors" are most important for assessing landslide susceptibility, they should quantify multicollinearity and assess whether seemingly unimportant variables are being masked by one or more collinear covariates. In particular, the authors should evaluate whether or not distance to the nearest road is masking other variables, as it essentially represents an interaction of multiple other causal factors.
Other major issues:
Line 12: I don't think a DEM should be called a "factor;" while slope (instability) is a causal factor of landslides, the DEM is not the same thing as the slope itself (the map is not the territory). I would suggest calling the DEM a "data source."
Lines 87-88: You wrote: "They concluded finer resolution has the effect on the increment of the extent of stable areas." I don't understand this sentence. Please try re-phrasing this sentence.
Lines 91-92: "statistically reliable" is unclear. Please be more specific about what Tian et al. concluded about the 10-m DEM. Did it lead to more accurate slope stability/ slope failure/ landslide risk assessments?
Lines 115-116: You write: "These techniques are selected based on their performances in that respective susceptibility modeling categories." This is hard to understand... I would suggest removing this sentence. I see that you have multiple citations here. Perhaps you meant to say that these techniques are the best-performing approaches for landslide susceptibility mapping. However, the way the sentence currently reads, it sounds like you are stating the criterion used in your own assessment (in this study).
Line 131 and Figure 1: Please expand the figure caption to include a description of what "Non-landslide locations" means in this figure. "Landslide locations" is clear, but any place that doesn't have a landslide is a non-landslide location, so what of the areas of the map that have no red dots nor green dots? My guess is that the green dots represent some random "pseudo-absence" points (to borrow an ecology term) as part of a sampling strategy for modeling, but you should make this clear in the figure caption.
Lines 136-138: You reference one landslide inventory that has 261 locations, then write that "from these locations" 168 were chosen... But also that they were "collected from" two different inventories (references 33 and 34). It's very confusing. Please clarify whether you filtered out some landslide locations from a larger collection and what the exact source is. Similarly, the following sentence, ("They employed...") seems out of place. If this inventory came from one study or multiple related studies, please introduce the study by name first, (e.g., "Rabby and Li prepared a nation-wide inventory of landslides using participatory field mapping...We selected 168 landslides from their inventory...").
Line 144: Please include a table in the main text listing the 15 causal factors and the data source for each. It's not very convenient for readers to have to consult the appendix (which will not accompany the published manuscript) to find out the causal factors of interest in this study.
Line 145: What is "slope plan"? And "profile curvatures"? I don't think these terms will be recognizable to most land-change scientists or remote sensing experts. Can you use different terms here?
Lines 151-153: How were the various data sources, with different native resolutions, resampled to a common 30-meter grid? In the table (to be added) that lists the causal factors, it would be a good idea to include the native resolutions of each data source.
Lines 154-155: Multivariate regression and random forest techniques can handle continuous data--why did you need to classify these continuous datasets? This is a poor choice for logistic regression and random forest. The random forest algorithm would generally do a much better job at discovering (in the data) a split at, e.g., 1000 m elevation than evaluating between prescribed (fixed) elevation bins. Classification may be necessary for the frequency ratio (FR) method only, so you should clarify whether you used classified values for the FR approach only or for all approaches.
Lines 156-226: Rather than having each of these in a separate section, you might have one section titled, e.g., "Selection of Covariates" (or "Selection of Factors") that justifies your decision to include each of these "causal factors."
Line 240: You write that there are 261 landslide locations used in this study. However, on Lines 136-138 you wrote that you selected 168 landslide locations. Please make sure you have the right number of landslide locations and that this number is consistent throughout the document.
Lines 245-246: You write: "Later we combined these data sets with the
training and validation landslide locations." This is confusing. You were just describing the training and validation datasets... So what is left to combine? Maybe this sentence should be deleted?
Lines 246-248: You write: "In total, we had 392 (196: landslide locations; 196: non-landslide locations) data points for training and 130 (65: landslide locations; 65: non-landslide locations) data points for testing the LR and RF models." Again, please check the numbers here... You just wrote that you had 194 non-landslide training locations, not 196. These numbers don't add up.
Lines 276-280: I think using a notation like "max(Rf_i)" or "min(Rf_i)" is more common than "Rfimax" or "Rfimin". It should also improve the presentation of "(Rfimax - Rfimin)min"
Lines 284 and 299: Note that you have two equations marked "Equation 5".
Line 299: Only one ellipsis (...) is really necessary here. I noticed you are using ellipses after every function and I just want to note this is probably note the style required by the journal.
Line 306-307: "We got statistically significant causal factors" is vague. You mean to say that every one of the causal factors was statistically significant? Please say as much if that is the case. Referencing a table of coefficients would be appropriate at this point.
Lines 307-308: You write: "Later, we multiplied the raster layers of these causal factors with the coefficients and summed up using Eq (5) in the R software environment." I think this is out of order... You would have to fit the model described by Equation 5 *before* obtaining any information about statistical significance. Perhaps delete the word "Later" and move this sentence to the beginning of the paragraph.
Line 354: I think it would be helpful to the reader if you defined the acronym "prediction rate" here, again.
Lines 368-375: This paragraph might fit better in the Discussion than in the Results.
Lines 383-400: While you provided sufficient reasoning for why distance to the road network might be correlated with landslide susceptibility, I think you should recognize that distance to the road network is also correlated with most of the other predictive features or "causal factors." While this multicollinearity would not affect the accuracy of a random forest prediction, it does substantially impact the variable importance metrics. I would suggest fitting a version of the random forest models where distance to the road network is excluded, so as to determine if any highly correlated factor is being masked by this variable. The same can be said for logistic regression; inclusion of highly collinear variables can lead to spuriously high p-values. You indicate, later in the paper that multicollinearity is a limitation of this study. However, it is very easily tested for (using variance inflation factors) and can be easily dealt with in this model evaluation study by producing sub-models where certain variables are excluded. Another approach would be factor analysis or a principal components analysis--not with the goal of explaining factors or principal components, but with the goals of examining what variables load onto the same factors and producing the best predictive model, even if the underlying relationships cannot be explained.
Figure 4: The acronyms "Pl" and "Pr" are used in the graphs but "PL" and "PC" are the closest matches described in the figure caption. Please check that the correct and consistent acronyms are used in both the figure and its caption.
Lines 611-613: You wrote: "Notably, the inclusion of common eight factors increases the prediction performance of other DEMs as well, but not as high as the SOB DEM." This is confusing. The "common eight factors" are the factors *not* derived from DEMs, so how do "increase[] the prediction performance of other DEMs"? It also sounds repetitive coming right after the previous sentence. Please revise both these sentences for clarity.
Minor grammatical or typographical edits:
Line 13: "causal factors often generated" should probably be "causal factors are often generated"
Line 14: "often relies on DEMs" should probably be "often rely on DEMs"
Line 39: "Identifying these causal factors are" should be "Identifying these causal factors is"
Line 98: "with fine spatial resolution not necessarily have" should probably be "with fine spatial resolution may not necessarily have"
Line 99: "course" should probably be "coarse"
Line 99: "upmost" should probably be "utmost" or another, simpler word like "urgent"
Line 100: "With this vain..." This doesn't make sense. Perhaps the author(s) meant, e.g., "Within this vein..." But I would suggest the author(s) avoid using English idioms altogether.
Line 102: "parts of the country encounters" should either be "part of the country encounters" or "parts of the country encounter"
Line 112: "This study aims compare" should be "This study aims to compare"
Line 331: I think "test the performance susceptibility models" was mean to be "test the performance of the susceptibility models"
Line 333: You wrote "The ‘area under the curve’ (AUC) of success rate curve" but I think this should just be "The 'area under the curve (AUC)...'"
Line 611: "the inclusions of common eight factors increase the prediction performance" needs to be corrected... I think it might read better as "the inclusion of eight other common factors increased the accuracy of predictions."
Author Response
Reviewer #4:
Response
Thank you for the comments and suggestions. These comments and suggestions have enabled us to improve the quality of our manuscript. We have accepted most of the corrections. Please check below the answers for each of the concerns about the manuscript.
- Currently, the Results section is very long and difficult to read because of the numerous acronyms, including the "model_data_DEM" acronyms. I understand the authors use these acronyms to simplify the description of a complicated cross-tabulation, but I think the Results section could be shortened by synthesizing the results across different data sources (e.g., SRTM, ASTER, etc.) and across different DEM sources. I think the authors have provided the reader with much more detail than is necessary. This hampers the reader's ability to understand what the key points of this study are. In particular, while the study reads as a model evaluation (i.e., which modeling approach provides the best results?), very little is said about how the different models (MFR, logistic regression, and random forest) compare to one another; instead, the authors focus on what combination of data inputs work best for each model. I should think they would want to make an overall recommendation about which model works best, as well. For example, they write in the Conclusion: "For the MFR model, we recommend complimenting the SOB DEM derived causal factors with other common causal factors (e.g. geology, land use/land cover)" (Lines 614-616). However, the study seems to indicate that the SOB DEM is generally unreliable ("Therefore, we argue that causal factors derived from the SOB DEM cannot explain the probability of landslides in the study area" Lines 613-614). Rather than recommending a specific approach that maximizes the utility of the SOB DEM, shouldn't the recommendation be not to use the SOB DEM in the first place? The authors do say this in the end (Line 623: "we suggest extreme caution before using SOB DEM for hilly areas in Bangladesh"). However, the authors could save space (and readers' time) by getting to the point much sooner, rather than describing the nuance of every possible model-data comparison. This is just one example of several issues throughout the paper where excessive description (and appendices) could be eliminated to make the paper more clear to readers.
Response: Thank you for pointing out these issues. We have revised the manuscript to remove unnecessary acronyms. We have also added a discussion section and reduced the size of result section. We did not compare three methods since our main aim was to compare the effects of DEMs on landslide susceptibility maps. In the discussion and conclusion section we have given the over all recommendation. Since three global DEM based landslide susceptibility maps did not have significant difference in performance therefore, we recommend using any one of them for landslide susceptibility mapping. We have recommended that researchers should not use SOB DEM because its quality is questionable. Please check lines 684-688.
- There are also a couple of substantial technical issues with the paper in its current form. First, I think it is a poor choice to classify the values of the causal factors used in logistic regression and random forest. While this classification is necessary for the frequency ratio approach, it only serves to artificially restrict the variance in the predictive features for logistic regression and random forest. I have seen others do this before in model evaluation studies, with the idea that it facilitates a "fair" comparison between models that require discrete inputs (e.g., frequency ratio) and those that don't (e.g., logistic regression, random forest). However, this is a dubious argument, because forcing models that can accept continuous data to use arbitrarily classified inputs is putting a severe handicap on the more flexible models. Conversely, those models that require discrete data ought to be evaluated as to whether or not they provide higher predictive accuracy given their strong assumptions.
Response: Thank you for pointing these issues. We classified the causal factors for modified frequency ratio model only. We did not use it for random forest and logistic regression model. We used the raw data.
- Second, the impact of multicollinearity on variable importance is not explored here. While it would not affect the predictive accuracy of the logistic regression and random forest models, it can substantially impact variable importance rankings or significance test. If the authors wish to make conclusions about which predictive features or "causal factors" are most important for assessing landslide susceptibility, they should quantify multicollinearity and assess whether seemingly unimportant variables are being masked by one or more collinear covariates. In particular, the authors should evaluate whether or not distance to the nearest road is masking other variables, as it essentially represents an interaction of multiple other causal factors.
Response: we have checked the multicollinearity issue. We did not find any highly correlated causal factors. Land use/ land cover and land use/land cover had higher dependency with each other compared to other causal factors. But tolerance and VIF was within the threshold so we included all the factors in the models. Please check lines 331-335.
- Line 12: I don't think a DEM should be called a "factor;" while slope (instability) is a causal factor of landslides, the DEM is not the same thing as the slope itself (the map is not the territory). I would suggest calling the DEM a "data source."
Response: thank you for raising this issue. We have accepted the suggestion.
5 Lines 87-88: You wrote: "They concluded finer resolution has the effect on the increment of the extent of stable areas." I don't understand this sentence. Please try re-phrasing this sentence.
Lines 91-92: "statistically reliable" is unclear. Please be more specific about what Tian et al. concluded about the 10-m DEM. Did it lead to more accurate slope stability/ slope failure/ landslide risk assessments?
Response: We have rewritten the sentence. Please check lines 85-88
- Lines 115-116: You write: "These techniques are selected based on their performances in that respective susceptibility modeling categories." This is hard to understand... I would suggest removing this sentence. I see that you have multiple citations here. Perhaps you meant to say that these techniques are the best-performing approaches for landslide susceptibility mapping. However, the way the sentence currently reads, it sounds like you are stating the criterion used in your own assessment (in this study).
Response: We have removed the sentence.
- Line 131 and Figure 1: Please expand the figure caption to include a description of what "Non-landslide locations" means in this figure. "Landslide locations" is clear, but any place that doesn't have a landslide is a non-landslide location, so what of the areas of the map that have no red dots nor green dots? My guess is that the green dots represent some random "pseudo-absence" points (to borrow an ecology term) as part of a sampling strategy for modeling, but you should make this clear in the figure caption.
Response: We have given a new caption to this image. We have used the word you suggested: pseudo-absence.
- Lines 136-138: You reference one landslide inventory that has 261 locations, then write that "from these locations" 168 were chosen... But also that they were "collected from" two different inventories (references 33 and 34). It's very confusing. Please clarify whether you filtered out some landslide locations from a larger collection and what the exact source is. Similarly, the following sentence, ("They employed...") seems out of place. If this inventory came from one study or multiple related studies, please introduce the study by name first, (e.g., "Rabby and Li prepared a nation-wide inventory of landslides using participatory field mapping...We selected 168 landslides from their inventory...").
Response: Thank you for pointing out this serious issue. We have selected 168 landslides published by Rabby and Li (2020). We have used Google Earth mapping to map 93 landslide locations proposed by the method of Rabby and Li (2019). Please check the rewritten Landslide Inventory section for detail.
- Line 144: Please include a table in the main text listing the 15 causal factors and the data source for each. It's not very convenient for readers to have to consult the appendix (which will not accompany the published manuscript) to find out the causal factors of interest in this study.
Lines 151-153: How were the various data sources, with different native resolutions, resampled to a common 30-meter grid? In the table (to be added) that lists the causal factors, it would be a good idea to include the native resolutions of each data source.
Response: Thank you for the suggestions. We have added the new table that addressed these issues. Please check Table 1.
- Line 145: What is "slope plan"? And "profile curvatures"? I don't think these terms will be recognizable to most land-change scientists or remote sensing experts. Can you use different terms here?
Response: Thank you for pointing out this issue. It was a spelling mistake. Slope plan should be slope. There were two curvatures used in this study: plan curvature and profile curvature. We have rewritten this part.
- Lines 154-155: Multivariate regression and random forest techniques can handle continuous data--why did you need to classify these continuous datasets? This is a poor choice for logistic regression and random forest. The random forest algorithm would generally do a much better job at discovering (in the data) a split at, e.g., 1000 m elevation than evaluating between prescribed (fixed) elevation bins. Classification may be necessary for the frequency ratio (FR) method only, so you should clarify whether you used classified values for the FR approach only or for all approaches.
Response: Thank you for pointing these issues. We classified the causal factors for modified frequency ratio model only. We did not use it for random forest and logistic regression model. We used the raw data.
- Lines 156-226: Rather than having each of these in a separate section, you might have one section titled, e.g., "Selection of Covariates" (or "Selection of Factors") that justifies your decision to include each of these "causal factors."
Response: Thank you for raising this issue. Since most of the landslide susceptibility mapping papers use this format, we have used it for our study.
- Line 240: You write that there are 261 landslide locations used in this study. However, on Lines 136-138 you wrote that you selected 168 landslide locations. Please make sure you have the right number of landslide locations and that this number is consistent throughout the document.
Response: Thank you for pointing out this important issue. We have revised it. Please check lines 239-247 for clarification.
- Lines 245-246: You write: "Later we combined these data sets with the
training and validation landslide locations." This is confusing. You were just describing the training and validation datasets... So what is left to combine? Maybe this sentence should be deleted?
Response: Thank you for pointing out this important issue. We have deleted it. Please check lines 239-247 for clarification.
- Lines 246-248: You write: "In total, we had 392 (196: landslide locations; 196: non-landslide locations) data points for training and 130 (65: landslide locations; 65: non-landslide locations) data points for testing the LR and RF models." Again, please check the numbers here... You just wrote that you had 194 non-landslide training locations, not 196. These numbers don't add up.
Response: Thank you for raising this issue. We have revised it.
- Lines 276-280: I think using a notation like "max(Rf_i)" or "min(Rf_i)" is more common than "Rfimax" or "Rfimin". It should also improve the presentation of "(Rfimax - Rfimin)min"
Lines 284 and 299: Note that you have two equations marked "Equation 5".
Line 299: Only one ellipsis (...) is really necessary here. I noticed you are using ellipses after every function and I just want to note this is probably note the style required by the journal.
Response: Thank you for your suggestion. We have accepted the notation that you suggested.
We have renumbered the equations.
We have revised the ellipsis.
- Line 306-307: "We got statistically significant causal factors" is vague. You mean to say that every one of the causal factors was statistically significant? Please say as much if that is the case. Referencing a table of coefficients would be appropriate at this point.
Response: We have revised these lines.
- Lines 307-308: You write: "Later, we multiplied the raster layers of these causal factors with the coefficients and summed up using Eq (5) in the R software environment." I think this is out of order... You would have to fit the model described by Equation 5 *before* obtaining any information about statistical significance. Perhaps delete the word "Later" and move this sentence to the beginning of the paragraph
Response: We have renumbered the equation.
- Line 354: I think it would be helpful to the reader if you defined the acronym "prediction rate" here, again.
Lines 368-375: This paragraph might fit better in the Discussion than in the Results.
Response: We have defined the acronym. We have added a discussion section and put the lines there.
- Lines 383-400: While you provided sufficient reasoning for why distance to the road network might be correlated with landslide susceptibility, I think you should recognize that distance to the road network is also correlated with most of the other predictive features or "causal factors." While this multicollinearity would not affect the accuracy of a random forest prediction, it does substantially impact the variable importance metrics. I would suggest fitting a version of the random forest models where distance to the road network is excluded, so as to determine if any highly correlated factor is being masked by this variable. The same can be said for logistic regression; inclusion of highly collinear variables can lead to spuriously high p-values. You indicate, later in the paper that multicollinearity is a limitation of this study. However, it is very easily tested for (using variance inflation factors) and can be easily dealt with in this model evaluation study by producing sub-models where certain variables are excluded. Another approach would be factor analysis or a principal components analysis--not with the goal of explaining factors or principal components, but with the goals of examining what variables load onto the same factors and producing the best predictive model, even if the underlying relationships cannot be explained.
Response: We have conducted the multicollinearity test and found no issue. We have deleted these lines in the conclusion.
- Figure 4: The acronyms "Pl" and "Pr" are used in the graphs but "PL" and "PC" are the closest matches described in the figure caption. Please check that the correct and consistent acronyms are used in both the figure and its caption.
Response: Thank you or raising this issue. We have revised the figure.
- Lines 611-613: You wrote: "Notably, the inclusion of common eight factors increases the prediction performance of other DEMs as well, but not as high as the SOB DEM." This is confusing. The "common eight factors" are the factors *not* derived from DEMs, so how do "increase[] the prediction performance of other DEMs"? It also sounds repetitive coming right after the previous sentence. Please revise both these sentences for clarity.
Response: Thank you for pointing out this issue in the conclusion section. We have revised the whole conclusion section. Therefore, we have got rid of these problems.
- Line 13: "causal factors often generated" should probably be "causal factors are often generated"
Line 14: "often relies on DEMs" should probably be "often rely on DEMs"
Line 39: "Identifying these causal factors are" should be "Identifying these causal factors is"
Line 98: "with fine spatial resolution not necessarily have" should probably be "with fine spatial resolution may not necessarily have"
Line 99: "course" should probably be "coarse"
Line 99: "upmost" should probably be "utmost" or another, simpler word like "urgent"
Line 100: "With this vain..." This doesn't make sense. Perhaps the author(s) meant, e.g., "Within this vein..." But I would suggest the author(s) avoid using English idioms altogether.
Line 102: "parts of the country encounters" should either be "part of the country encounters" or "parts of the country encounter"
Line 112: "This study aims compare" should be "This study aims to compare"
Line 331: I think "test the performance susceptibility models" was mean to be "test the performance of the susceptibility models"
Line 333: You wrote "The ‘area under the curve’ (AUC) of success rate curve" but I think this should just be "The 'area under the curve (AUC)...'"
Line 611: "the inclusions of common eight factors increase the prediction performance" needs to be corrected... I think it might read better as "the inclusion of eight other common factors increased the accuracy of predictions."
Response: We have accepted the suggestions.
Reviewer 5 Report
Dear authors,
I reviewed the paper entitled “Evaluating the effects of digital elevation models in landslide susceptibility mapping in Bangladesh” by Rabby et al.
The paper evaluates the performance of four different DEMs: ASTER GDEM, SRTM, ALOS PALSAR, and a local DEM of Bangladesh (SOB) for landslide susceptibility mapping in a hilly area of Bangladesh where frequent landslides occur every year. For that, the authors used three different models: a bivariate (modified frequency ratio), a multivariate (logistic regression), and a machine-learning (random forest) model, which have been deeply tested in the literature. Also, two scenarios have been defined in the study, one in which only DEM derived causal factors are considered and a second one in which apart from DEM other common factors are taken into account. The selection of these factors, apart from the methodology used and area of study, is very important. The results could differ depending on the considered number and type of causal factors. In addition, the quality of data and DEMs influence the results of the susceptibility maps as their performances depend on the area of study, not being extrapolated the results directly to other areas.
The paper is well written and good elaborated in general, quite long with a detailed analysis and description of the methodology. Sometimes, it is difficult to follow due to the amount of data, plots and information but it is not a bad point. The study is detailed and many variables are considered. The introduction provides enough background information and the objectives of the study are defined. In this sense, it should indicate that two different scenarios are considered in the study defining the causal factors to be included in each one. The data are described in detail and the methodology is clearly exposed.
The studies of landslide susceptibility, as said, depend on many factors, quality, methodology, and so on. According to the performed study in this area of Bangladesh the results show that the high accuracies in both scenarios are got using the SRTM DEM with the modified frequency ratio method. Also, ALOS DEM has a good performance with the other two methods getting the worst results for the three models and the two scenarios when the local SOB DEM is used. This is a DEM with a resolution of 25 m but it seems that its quality is poor. Are there similar studies where this SOB DEM is used and also obtained poorer outcomes?
The conclusions are supported by the results but the paper lacks of a “discussion” section where the results are compared with other studies in which similar causal factors are used and the performance of different DEMs is analyzed. I think this will enrich the paper.
Some minor points below.
Check the reference [69]. I did not find it cited in the text. Also check the reference 63 as it is not cited in the text. In this sense, check the reference in line 337 as it seems to be 63 instead of 53.
Line 19. Remove “and” before 12.5 m.
Line 115. Add “and” before “random forest”.
Line 116. Replace “that” with “those/these”.
Line 122. Replace square kilometer by IS symbol.
Line 172. For the aspect, it is said that ten classes are considered but in Table D1, for ASTER, there are nine classes and for the other three DEMs, there are ten classes being the “north” class repeated.
Equations in sections 3.2.5, 4.2, and 4.3 are of low quality. Use an equation editor. Also indicate how many classes have been considered for TWI and SPI according to Table D1.
Line 214. Six types of geologic formation are considered. However, in the Table D1 there are seven classes plus water and in Fig. 6 six classes plus water. Check it.
Line 225. Indicate the number of classes used for land use land cover changes as in Table D1 there are 10 classes plus a “no change” class and in Fig. 9 only 9 classes plus “no change”. Be consistent using a slash “/” in all the paper for “land use/land cover”.
Check Table 1 as the acronyms used for the last 10 models are incorrect.
Check section 4.2 as the numbers of equations are not correct from this section on (review all the paper). Equation number (2) is repeated and this changes everything.
Also check the variable names used in equations and in Table D1. Be consistent with capital or lower case letters.
Check the sentence in lines 273-275 as is not clear.
Line 411. Replace “was” with “were”.
Line 417. Replace “were” with “was”.
Figures 2 and 3. Remove “Figure” for each subplot and leave only letter and number.
Figure 4. Place “a” and “b” on the same location with respect to the subplots.
Line 534. Replace “6a, 6d” with “6a and 6d”.
Figure 5 caption. Express it clearly indicating which subplots are of each curve (also for Figure 6 caption). Check all the subplots in this figure and homogenize them. The location of the legend in the first one seems different to the others.
Figure A. The titles of the legend are excessively large. Add a title for every subplot in the same subplot. What is the reason for the change of resolution of subplot “l” in relation to “I, j, and k”? It is very coarse.
Figure B. Why is not this figure in color? Add a title for each subplot as indicated for Figure A. Also for Figure C. In subplot C.h is very difficult to appreciate the changes.
Table D1. Add in the caption the meaning of every variable in the header of this table.
It is difficult to read the paper because the order of the causal factors in the Table D1 is different to the order in which they appear in the text. So to improve the readability, be consistent and maintain the same order for the causal factors in this table as the order in which they appear in subsections 3.2.1 to 3.2.10.
In general, for all the tables repeat the header in every page that the table continues.
Table E1 caption. Remove a double “:”. Indicate in the caption the meaning of every variable in the header. What do 1 to 5 mean in this table?
Author Response
Reviewer #5:
Response
Thank you for the comments and suggestions. These comments and suggestions have enabled us to improve the quality of our manuscript. We have accepted most of the corrections. Please check below the answers for each of the concerns about the manuscript.
- The conclusions are supported by the results but the paper lacks a “discussion” section where the results are compared with other studies in which similar causal factors are used and the performance of different DEMs is analyzed. I think this will enrich the paper.
Response: thank you raising this important issue. We have added the discussion part.
- Check the reference [69]. I did not find it cited in the text. Also check the reference 63 as it is not cited in the text. In this sense, check the reference in line 337 as it seems to be 63 instead of 53.
Line 19. Remove “and” before 12.5 m.
Line 115. Add “and” before “random forest”.
Line 116. Replace “that” with “those/these”.
Line 122. Replace square kilometer by IS symbol.
Line 172. For the aspect, it is said that ten classes are considered but in Table D1, for ASTER, there are nine classes and for the other three DEMs, there are ten classes being the “north” class repeated.
Equations in sections 3.2.5, 4.2, and 4.3 are of low quality. Use an equation editor. Also indicate how many classes have been considered for TWI and SPI according to Table D1.
Line 214. Six types of geologic formation are considered. However, in the Table D1 there are seven classes plus water and in Fig. 6 six classes plus water. Check it.
Line 225. Indicate the number of classes used for land use land cover changes as in Table D1 there are 10 classes plus a “no change” class and in Fig. 9 only 9 classes plus “no change”. Be consistent using a slash “/” in all the paper for “land use/land cover”.
Check Table 1 as the acronyms used for the last 10 models are incorrect.
Check section 4.2 as the numbers of equations are not correct from this section on (review all the paper). Equation number (2) is repeated and this changes everything.
Also check the variable names used in equations and in Table D1. Be consistent with capital or lower case letters.
Check the sentence in lines 273-275 as is not clear.
Line 411. Replace “was” with “were”.
Line 417. Replace “were” with “was”.
Figures 2 and 3. Remove “Figure” for each subplot and leave only letter and number.
Figure 4. Place “a” and “b” on the same location with respect to the subplots.
Line 534. Replace “6a, 6d” with “6a and 6d”.
Figure 5 caption. Express it clearly indicating which subplots are of each curve (also for Figure 6 caption). Check all the subplots in this figure and homogenize them. The location of the legend in the first one seems different to the others.
Figure A. The titles of the legend are excessively large. Add a title for every subplot in the same subplot. What is the reason for the change of resolution of subplot “l” in relation to “I, j, and k”? It is very coarse.
Figure B. Why is not this figure in color? Add a title for each subplot as indicated for Figure A. Also for Figure C. In subplot C.h is very difficult to appreciate the changes.
Table D1. Add in the caption the meaning of every variable in the header of this table.
It is difficult to read the paper because the order of the causal factors in the Table D1 is different to the order in which they appear in the text. So to improve the readability, be consistent and maintain the same order for the causal factors in this table as the order in which they appear in subsections 3.2.1 to 3.2.10.
In general, for all the tables repeat the header in every page that the table continues.
Table E1 caption. Remove a double “:”. Indicate in the caption the meaning of every variable in the header. What do 1 to 5 mean in this table?
Response; Thank you for pointing out these problems. We have accepted the corrections and revised the manuscript to address these issues.
Round 2
Reviewer 1 Report
This paper is much improved. Here are a few more comments for the authors:
Table 1 is referenced in the text in line 159 (page 6) but is located on line 243 (page 8). Please move this table closer to line 159.
Line 292: the word "indicates" is misspelled as "inictaes".
Lines 381-382: The term 'prediction rates' is defined as PR in line 381, so you can delete the term 'prediction rates' and just use PR in line 382.
Figures 2 and 3 are MUCH better. Thank you.
Line 594: Formatting needs to be addressed (this is a full blank page) here and a few other locations in the paper, but this may be a product of editing and markups.
Nice work and well done.
Author Response
Reviewer #1:
Response
Thank you for the comments and suggestions. These comments and suggestions have enabled us to improve the quality of our manuscript. We have accepted most of the corrections. Please check below the answers for each of the concerns about the manuscript.
- Table 1 is referenced in the text in line 159 (page 6) but is located on line 243 (page 8). Please move this table closer to line 159.
Response: Thank you for your advice. We have moved the table. Please check line 155. We have first mentioned Table 1 in Line 143. Therefore, we moved the table in line 155.
- Line 292: the word "indicates" is misspelled as "inictaes".
Response: Thank you for pointing out this issue. We have revised the manuscript to check this sort of spelling error.
- Line 594: Formatting needs to be addressed (this is a full blank page) here and a few other locations in the paper, but this may be a product of editing and markups.
Response: Thank you for pointing out this issue. We have revised the whole manuscript to get rid of this problem.
Reviewer 2 Report
The authors have addressed almost all the reviewers' comments, and the manuscript in its current version is improved compared to the original.
I have no further comments, and the revised manuscript can be accepted.
Author Response
Reviewer #2:
Response
Thank you for the comments and suggestions. These comments and suggestions have enabled us to improve the quality of our manuscript.